# Competing scaffolding proteins determine capsid size during mobilization of *Staphylococcus aureus* pathogenicity islands

Altaira D Dearborn[1†], Erin A Wall[2†‡], James L Kizziah[3†], Laura Klenow[2], Laura K Parker[2,3], Keith A Manning[3], Michael S Spilman[4], John M Spear[5§], Gail E Christie[2], Terje Dokland[3*]

[1]Protein Expression Laboratory, National Institute of Arthritis and Musculoskeletal and Skin Diseases, National Institutes of Health, Bethesda, United States; [2]Department of Microbiology and Immunology, Virginia Commonwealth University, Richmond, United States; [3]Department of Microbiology, University of Alabama, Birmingham, United States; [4]Direct Electron, San Diego, United States; [5]Biological Science Imaging Resource, Florida State University, Tallahassee, United States

**\*For correspondence:** dokland@uab.edu

[†]These authors contributed equally to this work

**Present address:** [‡]Laboratory of Molecular Biology, Center for Cancer Research, National Cancer Institute, Bethesda, United States; [§]Thermo Fisher Scientific, Hillsboro, United States

## Abstract

*Staphylococcus aureus* pathogenicity islands (SaPIs), such as SaPI1, exploit specific helper bacteriophages, like 80α, for their high frequency mobilization, a process termed 'molecular piracy'. SaPI1 redirects the helper's assembly pathway to form small capsids that can only accommodate the smaller SaPI1 genome, but not a complete phage genome. SaPI1 encodes two proteins, CpmA and CpmB, that are responsible for this size redirection. We have determined the structures of the 80α and SaPI1 procapsids to near-atomic resolution by cryo-electron microscopy, and show that CpmB competes with the 80α scaffolding protein (SP) for a binding site on the capsid protein (CP), and works by altering the angle between capsomers. We probed these interactions genetically and identified second-site suppressors of lethal mutations in SP. Our structures show, for the first time, the detailed interactions between SP and CP in a bacteriophage, providing unique insights into macromolecular assembly processes.

DOI: https://doi.org/10.7554/eLife.30822.001

## Introduction

*Staphylococcus aureus* is an opportunistic pathogen sometimes associated with serious skin and soft tissue infections in humans and animals (*Lowy, 1998*). *S. aureus* encodes a large number of virulence factors that allow it to adapt to a variety of niches and to evade both the innate and adaptive immune system (*Archer, 1998*). The emergence of *S. aureus* strains that are resistant to most antibiotics in common use has become a significant public health concern (*Deurenberg et al., 2007*; *DeLeo et al., 2010*).

Most virulence factors in *S. aureus* are encoded on mobile genetic elements (MGEs) like bacteriophages, plasmids and genomic islands (*Lindsay, 2014*). *S. aureus* pathogenicity islands (SaPIs) are a family of genomic islands that encode a range of virulence factors, including toxins, adhesins, coagulation factors and immunomodulatory factors (*Diep et al., 2006*; *Novick et al., 2010*; *Viana et al., 2010*; *Novick and Ram, 2016*). Many SaPIs encode superantigen toxins, which hyperactivate the immune system by bypassing the normal antigen presentation mechanism (*Alouf and Müller-Alouf, 2003*). For example, SaPI1, SaPI2 and SaPIbov1 carry the *tst* gene, which encodes the toxic shock syndrome toxin (TSST-1) that is the causative agent of toxic shock, a frequently fatal condition

(*Kulhankova et al., 2014*). SaPI1 also includes *sel* and *seq*, encoding staphylococcal enterotoxins L and Q, respectively (*Lindsay et al., 1998*).

Transduction by phages is the main mode of horizontal transmission of MGEs in *S. aureus*. SaPIs engage in a kind of 'molecular piracy', in which they co-opt the structural proteins of a 'helper' bacteriophage for their own genome encapsidation and high-frequency transfer (*Christie and Dokland, 2012*). This arrangement differs from typical satellite viruses, which encode their own structural proteins, but depend on other viruses for other functions, and is far more efficient than generalized transduction. Bacteriophage 80α is a temperate siphovirus that can act as helper for several SaPIs, including SaPI1, SaPI2 and SaPIbov1 (*Christie et al., 2010*). 80α has a 63 nm diameter *T* = 7 icosahedral capsid and a 190 nm long, flexuous tail capped by an elaborate baseplate (*Spilman et al., 2011*). Like other tailed, double-stranded (ds) DNA bacteriophages (order Caudovirales), the 80α capsid protein (CP, 324 residues) is first assembled into an empty procapsid, requiring a scaffolding protein (SP, 206 residues), which acts as a chaperone for the assembly process. The 80α procapsid is subsequently packaged with DNA by a headful mechanism using the phage-encoded terminase, consisting of a small (TerS) and a large (TerL) subunit (*Feiss and Rao, 2012*). Tails, which are assembled independently, are added to the completed, DNA-filled capsids.

SaPIs are normally stably integrated into the host genome through expression of their Stl repressors (*Ubeda et al., 2008*). The Stl repressor detects the presence of a helper phage by interacting with a phage early lytic gene product, such as 80α Sri or the Dut dUTPase (*Tormo-Más et al., 2010*; *Hill and Dokland, 2016*), leading to derepression, excision and replication of the SaPI, followed by packaging into transducing particles made by phage-encoded proteins (*Figure 1*). This strategy allows the SaPI to escape the phage-induced cell lysis and instead spread horizontally through the bacterial population.

SaPIs interfere with the multiplication of their helpers in many ways, including the expression of a TerS subunit that specifically recognizes SaPI DNA (*Figure 1*), and proteins that sequester the phage-encoded TerS protein (*Christie and Dokland, 2012*; *Novick and Ram, 2016*). Many SaPIs also cause the redirection of the helper phage assembly pathway to form capsids that are smaller than those normally formed by the phage and are therefore unable to package complete phage genomes (*Christie and Dokland, 2012*). For SaPI1, we previously demonstrated that this size redirection depended on the SaPI1-encoded proteins CpmA and CpmB (*Poliakov et al., 2008*; *Dearborn et al., 2011*; *Damle et al., 2012*), leading to the formation of a 45 nm diameter capsid with *T* = 4 icosahedral symmetry (*Dearborn et al., 2011*) (*Figure 1*). We determined the NMR structure of N-terminal domain of CpmB, a dimeric, α-helical 72-residue protein (*Dearborn et al., 2011*) reminiscent of the scaffolding protein of bacteriophage φ29 (*Morais et al., 2003*). Reconstructions of SaPI1 procapsids revealed protrusions on the inside of the shell that we presumed to correspond to the CpmB dimers (*Dearborn et al., 2011*). SP and CpmB share a homologous RIIK motif near their C-termini, leading us to hypothesize that the two proteins might interact with CP in a similar fashion (*Dearborn et al., 2011*).

Our previous reconstructions of 80α and SaPI1 capsids were generated from data collected on film with our in-house Tecnai F20 microscope, and were limited in resolution to 9–10 Å (*Dearborn et al., 2011*; *Spilman et al., 2011*). This allowed a rough homology model for CP to be built; however, many regions were poorly resolved or ambiguous, and the SP was not observed in the density maps. Here, we present the icosahedrally symmetric structures of 80α and SaPI1 procapsids at near-atomic resolution, allowing the CP and part of the SP and CpmB to be accurately modeled into the density. Our structures show that CpmB affects the shell curvature around the threefold symmetry axes and binds to CP using the same interactions as the cognate SP. As a consequence, the phage is prevented from escaping the CpmB-induced capsid size redirection by mutating its CP. This study has shown, for the first time, the detailed interactions between a scaffolding protein and a capsid, and provides new insights into the assembly and size determination process for viruses and other macromolecular complexes.

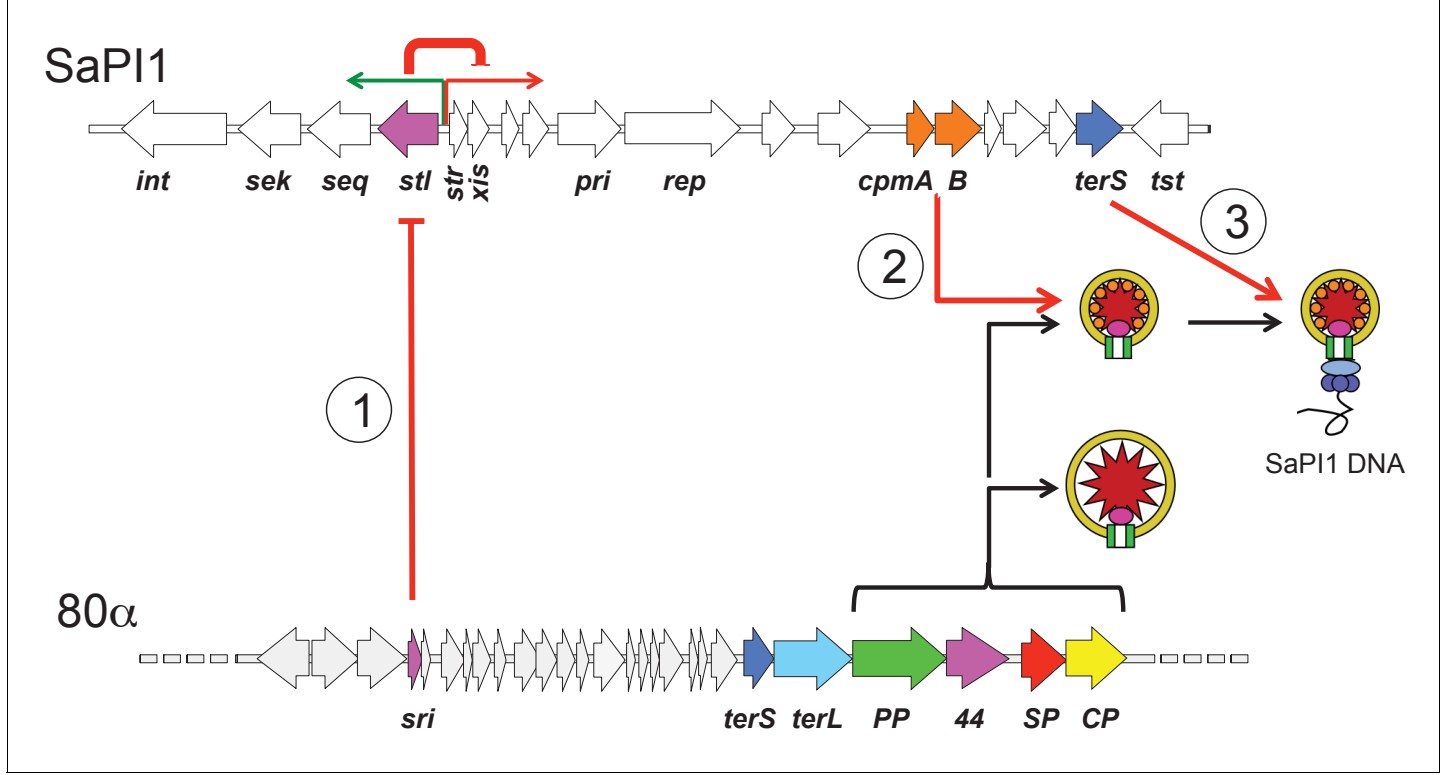

**Figure 1.** Schematic overview of SaPI1's molecular piracy of 80α. Only part of the 80α genome is shown for clarity, and relevant genes are labeled: *sri*, SaPI1 derepressor; *terS*, terminase small subunit; *terL*, terminase large subunit; *PP*, portal protein; *44*, minor capsid protein gp44; *SP*, scaffolding protein; *CP*, major capsid protein. First, the 80α-encoded protein Sri interacts with the SaPI1 master repressor Stl (1), causing derepression of the rightwards operon that includes the transcriptional activator *str*, the excisionase *xis* and the replicase (*pri* and *rep*). Second, the SaPI1-encoded proteins CpmA and CpmB re-direct the capsid assembly pathway to form small capsids (2). Finally, the SaPI1-encoded TerS interacts with 80α-encoded TerL to cause specific packaging of SaPI1 DNA (3). Also labeled are the SaPI1 integrase (*int*), genes encoding staphylococcal enterotoxins K (*sek*) and Q (*seq*), and the toxic shock syndrome toxin (*tst*) gene.

DOI: https://doi.org/10.7554/eLife.30822.002

## Results

### Structures of 80α and SaPI1 procapsids

Cryo-electron microscopy (EM) images of 80α and SaPI1 procapsids were collected on an FEI Titan Krios electron microscope equipped with a DE-20 direct electron detector (*Figure 2A,B*). The 80α and SaPI1 procapsid structures were solved to 3.8 Å and 3.7 Å resolution (FSC$_{0.143}$; *Figure 2C,D*) from 10,527 and 14,087 particle images, respectively, using icosahedral reconstruction with *jspr* (*Guo and Jiang, 2014*). The reconstructions show the expected $T = 7$ and $T = 4$ architectures of the 80α and SaPI1 procapsids, respectively, made up of hexamers and pentamers of CP subunits (*Figure 3A,B*). In these maps, the protein backbones could be traced through most of the density with confidence (*Figure 3—figure supplement 1A,B*). While many side chains are not fully defined at this resolution, bulky side chains could be identified sufficiently well to allow atomic models to be built for residues 26–309 of CP in both 80α and SaPI1, and for parts of SP and CpmB (*Figure 3—figure supplement 1C–F*). The models were refined in REFMAC5 and showed good correspondence with the data up to the resolution of the map (*Figure 2C,D*; *Figure 3—figure supplement 2A,B*). Some regions in the map—mainly corresponding to the N- and C-termini of CP—had worse resolution than the rest of the structure, presumably due to structural heterogeneity or disorder (*Figure 3—figure supplement 2C,D*), and could not be modeled unambiguously. The pentamer density generally had lower resolution than the hexamers, perhaps due to unequal packing of subunits at the fivefold vertex, and averaging with the dodecameric portal that occupies one of the twelve vertices.

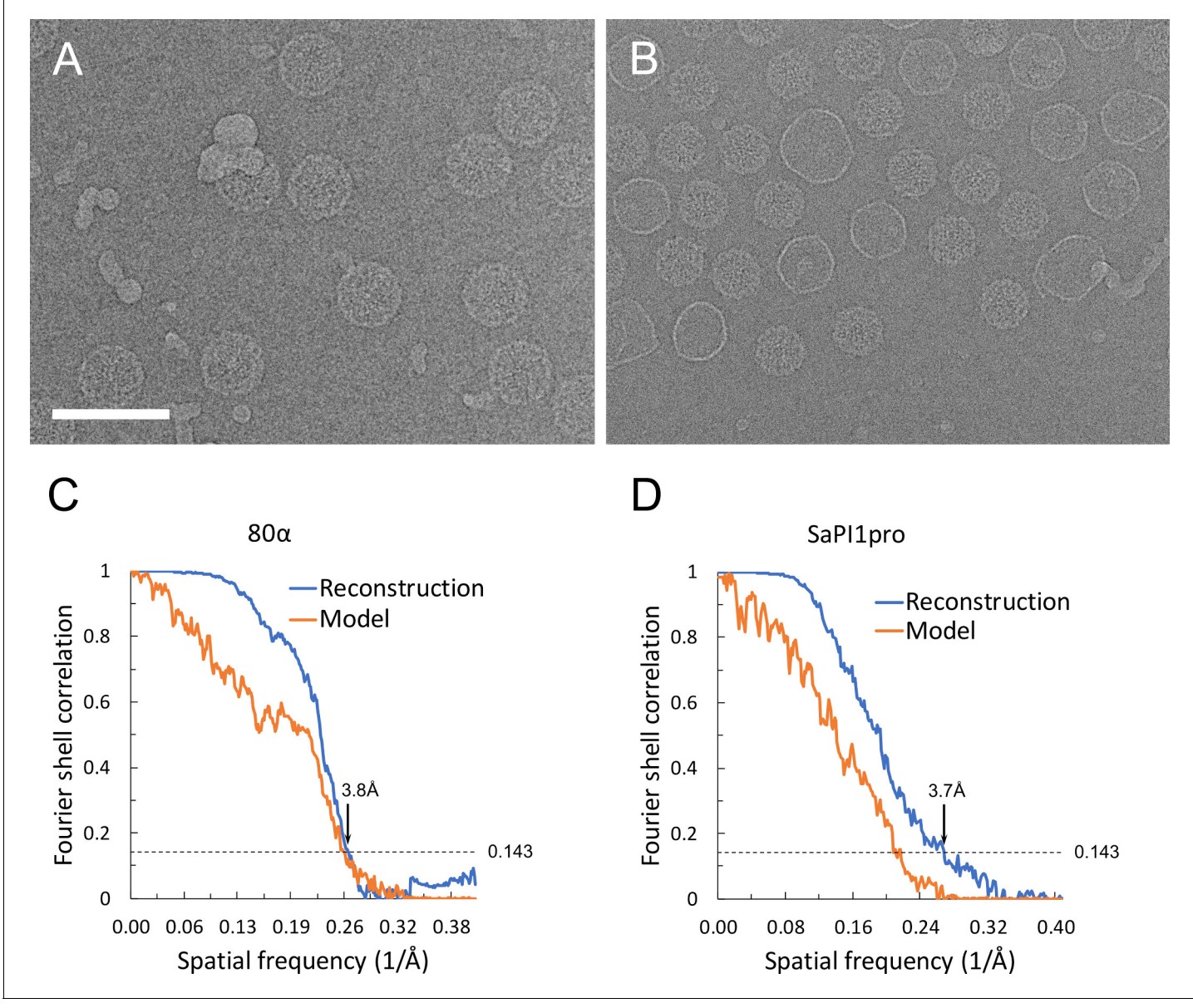

**Figure 2.** Cryo-EM data. (**A, B**) Electron micrographs of 80α (**A**) and SaPI1 (**B**) procapsids. Scale bar, 100 nm. (**C, D**) Fourier Shell Correlation (FSC) curves between maps calculated from the two half datasets (blue) and between the model and the map (orange) for 80α (**A**) and SaPI1 (**B**). The resolution limits at 0.143 cutoff (dashed line) are indicated. The discrepancy between the model and the map for SaPI1 is likely due to the disordered and unmodeled portions of the CpmB protrusions.

DOI: https://doi.org/10.7554/eLife.30822.003

The CP models show the expected HK97-like fold common to the Caudovirales, divided into A- and P-domains, with an extended E-loop and an N-arm that folds along the inside of the P-domain (*Figure 3C,D*). The long 'spine helix' (α3) in the P-domain is a characteristic feature of the HK97 fold. In 80α, this helix is kinked at residue P132 (*Spilman et al., 2011*). Residues 263–285 form an insertion into the P-domain called the P-loop. The P-loop and most of the E-loop were not resolved in the previous low-resolution maps (*Dearborn et al., 2011*; *Spilman et al., 2011*), but could be modeled with confidence in the new maps. In addition, the C-terminal part of SP could be identified, and was modeled de novo into the density (see below). In the SaPI1 reconstruction, internal densities corresponding to the previously described CpmB dimers (*Dearborn et al., 2011*) were clearly discernible (*Figure 3E*), but noisy, and only the C-terminal parts closest to the capsid were

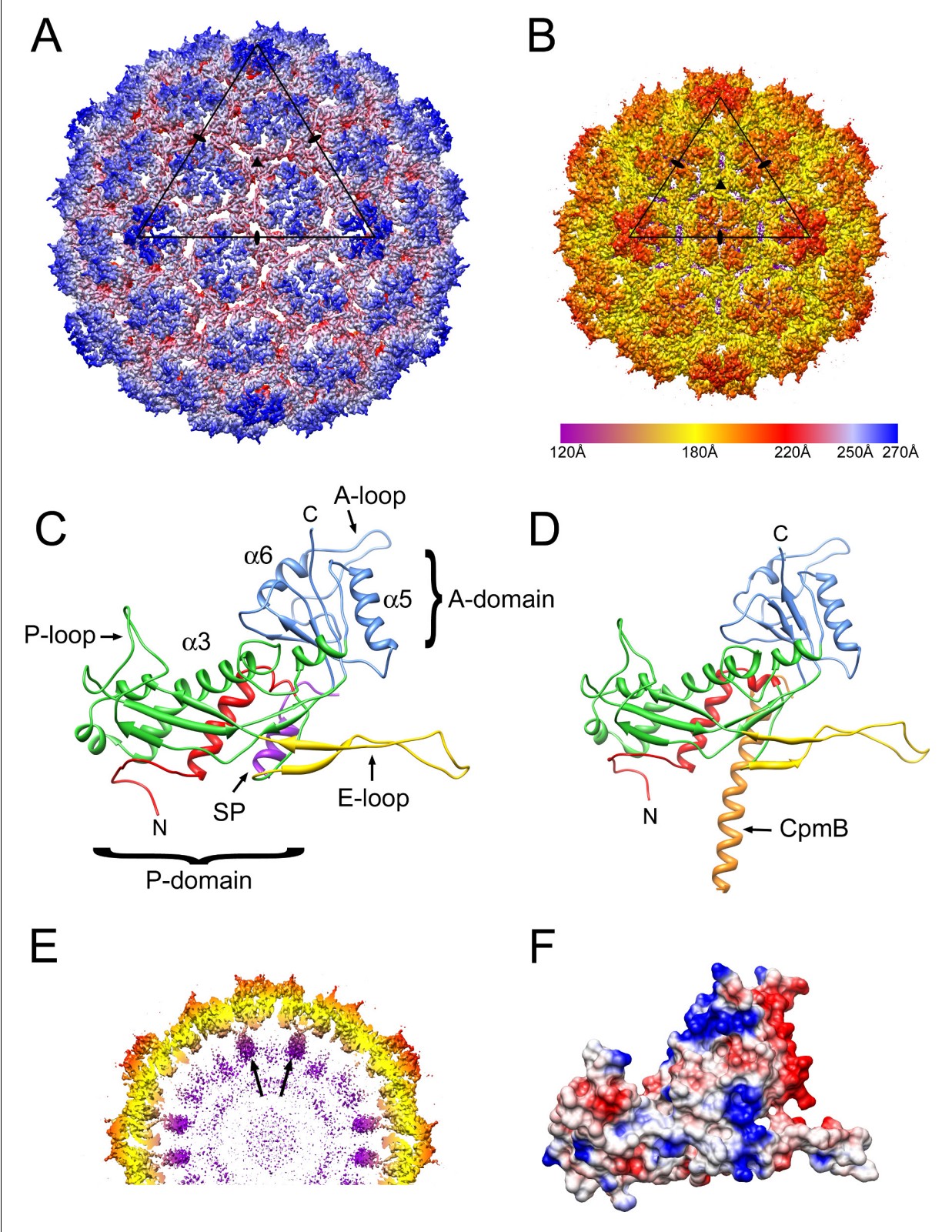

**Figure 3.** Reconstruction of 80α and SaPI1 procapsids. (**A, B**) Isosurface representation of the 80α (**A**) and SaPI1 (**B**) reconstructions, colored radially on an absolute scale (color bar). The triangles represent one icosahedral face, delimited by three fivefold axes. Icosahedral twofold and threefold symmetry axes are indicated by ovals and triangles, respectively. (**C, D**) Ribbon representations of the C subunits of the 80α (**C**) and SaPI1 (**D**) atomic models. N-arm, red; P domain, green; E-loop, yellow; A-domain, blue. SP (in 80α), purple; CpmB (in SaPI1), orange. Locations of α-helices α3 (the spine helix),

*Figure 3 continued on next page*

*Figure 3 continued*

α5 and α6, and the A-loop are also indicated in (**C**). (**D**) Isosurface representation of a 30 Å thick slice through the SaPI1 procapsid map, showing the protruding domains on the inside of the shell (arrows), colored radially on the same scale as in (**B**). (**G**) Electrostatic surface of the 80α CP subunit C, colored according to charge (blue, most positive; red, most negative).

DOI: https://doi.org/10.7554/eLife.30822.004

The following figure supplements are available for figure 3:

**Figure supplement 1.** Electron density and model.

DOI: https://doi.org/10.7554/eLife.30822.005

**Figure supplement 2.** Data analysis.

DOI: https://doi.org/10.7554/eLife.30822.006

interpretable in terms of the molecular structure of the protein (*Figure 3D*).An electrostatic surface representation of one of the CP subunits shows a characteristic distribution of positive and negative charges on opposite sides of the A-domain (*Figure 3F*).

There are seven CP subunits (A–G) in the asymmetric unit in the $T$ = 7 80α procapsid structure, which are organized into $A_5$ pentamers and BCDEFG hexamers (*Figure 4A*). The seven subunits can be superimposed with overall root-mean-square deviations (RMSD) ranging from 1.0 Å to 3.3 Å, with most of the differences localized to the E-loops (*Figure 4A*; *Table 1*). In the $T$ = 4 SaPI1 procapsid, there are four subunits, organized as $A_5$ pentamers and (BCD)$_2$ hexamers (*Figure 4B*), which superimpose with RMSD = 1.5–2.9 Å (*Figure 4B*; *Table 1*). These hexamers sit on the icosahedral twofold axes and thus have strict twofold symmetry. Strikingly, this twofold symmetry is retained in the 80α hexamers although no symmetry is imposed. Thus, the most similar subunits in the 80α capsid are those pairs related across this quasi-twofold symmetry axis (B–E, C–F and D–G), with RMSD values between 1.0–1.5 Å (*Table 1*). The entire hexamer and pentamer assemblies are very similar between 80α and SaPI1 (*Figure 4C,D*), and corresponding subunits in 80α and SaPI1 superimpose with RMSD ≤1.5 Å (*Table 1*).

Interactions between subunits within the hexamers and pentamers are mediated primarily via the E-loops and A-domains. In the A-domains, interactions involve several charged residues in the α5 and α6 helices and in the A-loops (*Figure 5A,B*). The hexamers have a distinct 'skew' along a plane that separates the CDE subunits from FGB, leading to distinct differences in inter-subunit interactions in the A domains (*Figure 5A*). This skew is also accommodated by conformational differences in the E-loops, which segregate into two distinct classes related by a 20° rotation (*Figure 4A,B*). The 'up' conformation includes the two subunits whose E-loops reach across this skew plane (C and F) as well as the pentamer subunit (A), while the other subunits are in the 'down' conformation. In is unclear whether the E-loops simply conform to the skewed hexamer geometry, or themselves induce the skew in the hexamers. The E-loop forms a β-hairpin structure that contains several aromatic residues and wraps around the P-domain of the adjacent subunit, where it engages with the P-loop via a short β-sheet (*Figure 5C*). The contact surface between the E-loop and the adjacent subunit's P domain is extensive, covering an area of ≈1000 Å$^2$ or ≈6% of the subunit surface.

To complete the shell assembly, the capsomers are tied together via trivalent interactions, of which there are three types in the 80α capsid (ABG, CDF and EEE; *Figure 4A*) and two in SaPI1 (ABD and CCC; *Figure 4B*). These interactions are mediated by several residues in the P-loops, which come together to form a trefoil structure that also involves residues at the tip of the E-loop of the adjacent subunit (*Figure 5D*). An additional contact is made by two quasi-twofold related W98 residues in the P domains that engage in parallel-displaced aromatic stacking (*Figure 5D*).

To define the differences between the 80α and SaPI1 shells, each capsomer was represented by a plane drawn between equivalent atoms in the constituent (five or six) subunits, and the interior, dihedral angles between planes were calculated. There are four such angles in the 80α capsid (α, β, γ and δ) and two in SaPI1 (α and β) (*Figure 6A,B*). These angles were consistently greater in 80α capsids compared to SaPI1 (*Figure 6C*). The angle that relates hexamers around the icosahedral threefold axis (α) differed more (13.1°) than that relating hexamers to pentamers (β). Such differences, propagated throughout the lattice, lead to the overall greater curvature that defines the small SaPI1 capsid (*Figure 6D*).

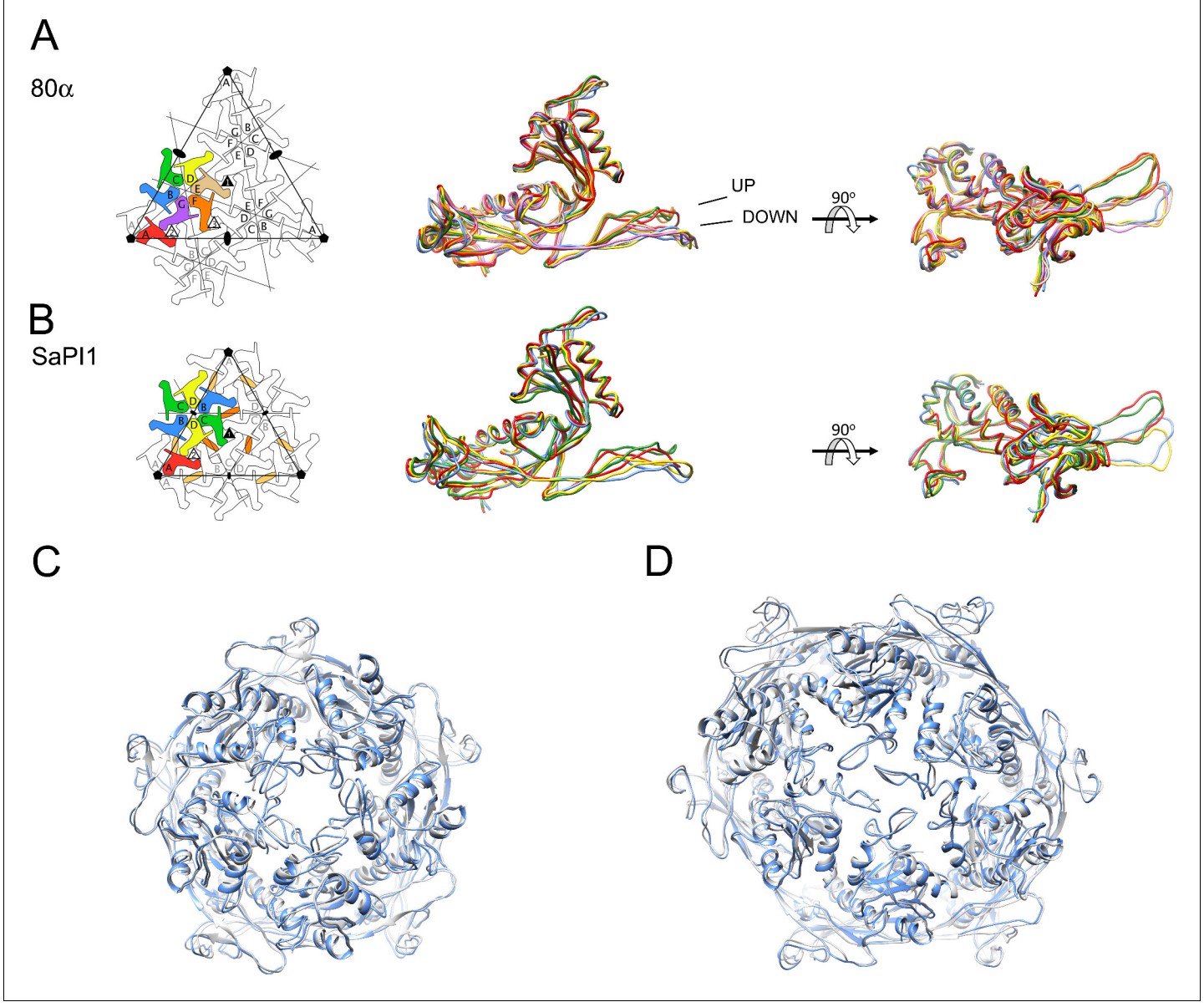

**Figure 4.** Comparison of CP subunits. (A, B) Superposition of the backbones of the seven subunits of 80α (A) and the four subunits of SaPI1 (B). Panels on the right are rotated by 90°. The two distinct conformations of the E-loop ('UP' and 'DOWN') are indicated on the 80α superposition. The subunits are colored according to the schematic diagrams on the left (A, red; B, blue; C, green; D, yellow; E, tan; F, orange; G, purple). The orange ovals in (B) represent the CpmB dimers. (C, D) Superpositions of the pentamers (C) and hexamers (D) of 80α (gray) and SaPI1 (blue), in ribbon representation.

DOI: https://doi.org/10.7554/eLife.30822.007

## Interactions between scaffolding and capsid proteins

The 80α procapsid map had a density next to N-arm helix α1 (*Figure 7A*) that was identified as the 16 C-terminal residues of SP, which were built de novo as an 11-residue amphipathic α-helix followed by a 5-residue 'hook'. The lower density of this feature is indicative of low occupancy, consistent with the previously estimated CP:SP molar ratio of 2:1 (*Poliakov et al., 2008*). The hook includes the conserved RIIK motif that was previously hypothesized to interact with CP (*Dearborn et al., 2011*). The RIIK motif points into the hydrophobic cleft between the A and P domains, where R202, I203 and I204 make numerous contacts with the CP (*Figure 7B*; *Table 2*). In the preceding α-helix, L194, I197, A198 and K201 make contacts predominantly with the N-arm α-helix of CP (*Figure 7B*; *Table 2*). In subunits C and F only, residues Q192, R199 and N206 contact

**Table 1.** Root-mean square deviation (RMSD) values (in Å) between subunit pairs within 80α, within SaPI1 and between 80α and SaPI1.

Values for 'unpruned' structures include all atoms when determining the best fit. Values for 'pruned' structures only include the best matching residue pairs. Pruning was done in UCSF Chimera using standard parameters. For 80α, pairs of subunits related across the quasi-twofold axis are highlighted in green. Comparison of subunits for which the E-loops are in the 'UP' conformation are highlighted in blue.

**Root mean square deviation between CP subunits**

| | | Pruned | | Unpruned | |
|---|---|---|---|---|---|
| | | # res. | RMSD (Å) | # res. | RMSD (Å) |
| **80α procapsid** | | | | | |
| subA | subB | 215 | 0.986 | 284 | 2.927 |
| subA | subC | 250 | 0.897 | 284 | 1.521 |
| subA | subD | 239 | 0.943 | 284 | 2.902 |
| subA | subE | 218 | 0.989 | 284 | 2.900 |
| subA | subF | 252 | 0.899 | 284 | 1.533 |
| subA | subG | 239 | 1.024 | 284 | 2.878 |
| subB | subC | 224 | 0.881 | 284 | 3.072 |
| subB | subD | 229 | 0.955 | 284 | 1.901 |
| subB | subE | 251 | 0.727 | 284 | 1.506 |
| subB | subF | 219 | 0.907 | 284 | 3.244 |
| subB | subG | 232 | 0.909 | 284 | 1.686 |
| subC | subD | 241 | 0.853 | 284 | 3.002 |
| subC | subE | 225 | 0.923 | 284 | 3.108 |
| subC | subF | 271 | 0.764 | 284 | 0.987 |
| subC | subG | 237 | 0.860 | 284 | 2.974 |
| subD | subE | 231 | 1.001 | 284 | 1.715 |
| subD | subF | 246 | 0.781 | 284 | 3.195 |
| subD | subG | 262 | 0.704 | 284 | 1.302 |
| subE | subF | 220 | 0.979 | 284 | 3.257 |
| subE | subG | 236 | 0.981 | 284 | 1.654 |
| subF | subG | 242 | 0.847 | 284 | 3.104 |
| **SaPI1 procapsid** | | | | | |
| subA | subB | 218 | 1.047 | 284 | 2.656 |
| subA | subC | 247 | 1.092 | 284 | 1.572 |
| subA | subD | 223 | 1.108 | 284 | 2.862 |
| subB | subC | 220 | 1.040 | 284 | 2.865 |
| subB | subD | 231 | 1.040 | 284 | 1.633 |
| subC | subD | 242 | 0.921 | 284 | 2.669 |
| **SaPI1 vs. 80α** | | | | | |
| **SaPI1** | **80α** | | | | |
| subA | subA | 259 | 0.920 | 284 | 1.495 |
| subB | subB | 250 | 0.926 | 284 | 1.443 |
| subB | subE | 257 | 0.823 | 284 | 1.380 |
| subC | subC | 268 | 0.935 | 284 | 1.165 |
| subC | subF | 268 | 0.997 | 284 | 1.190 |
| subD | subD | 247 | 0.948 | 284 | 1.536 |
| subD | subG | 262 | 0.939 | 284 | 1.170 |

DOI: https://doi.org/10.7554/eLife.30822.008

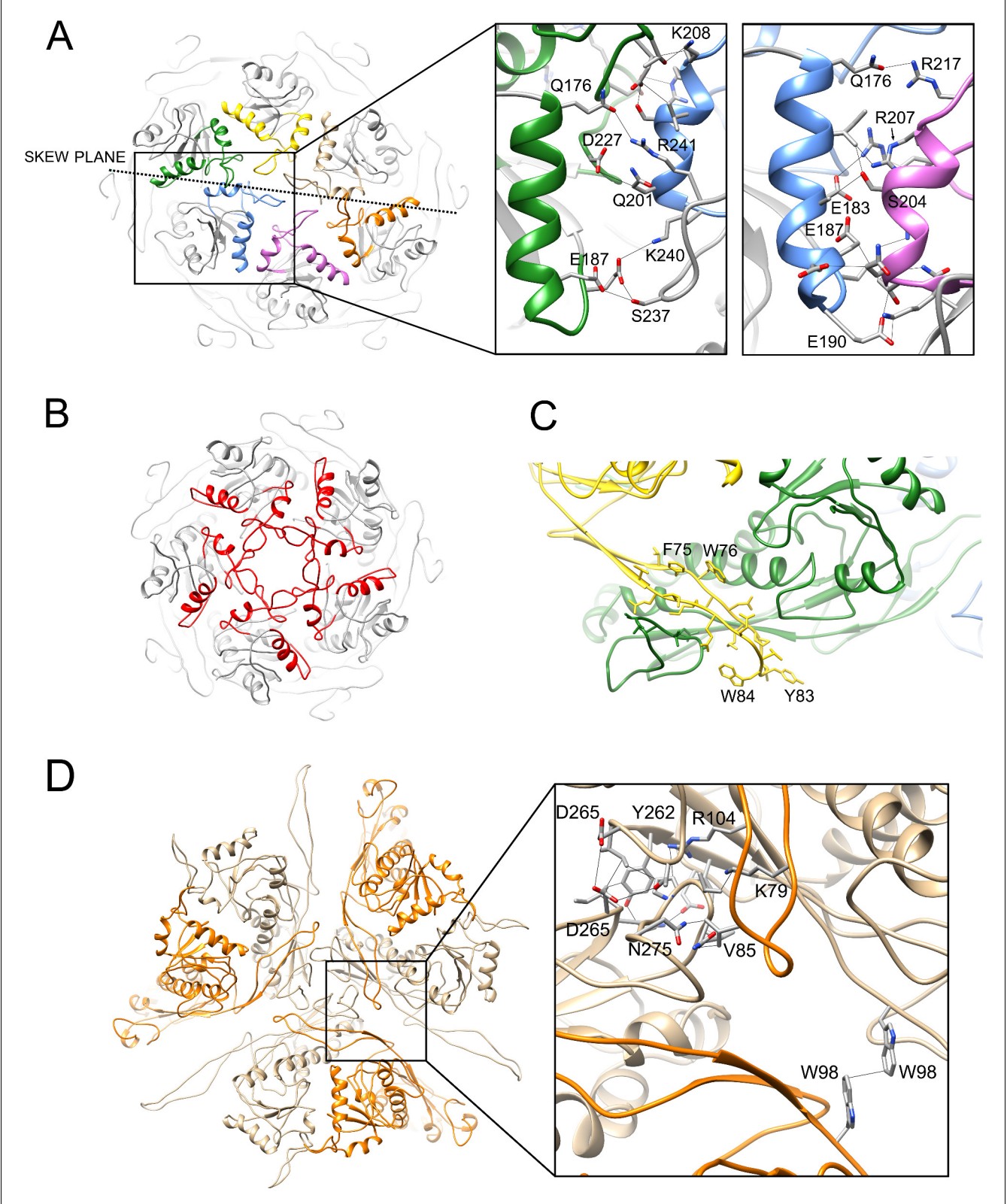

**Figure 5.** Interactions between capsid proteins. (**A, B**) Ribbon representation of the 80α pentamer (**A**) and hexamer (**B**), showing the detailed interactions between the A-domains. Helices α5 and α6 and the A-loop are colored according to the color scheme in *Figure 4A*. The skew plane between B-C and E-F subunits is indicated by the dashed line. Pertinent residues are shown in stick representation and labeled in the expanded view on the right, showing the two distinct types of A-domain interactions. (**C**) Detail of the 80α model, showing the interactions between the E-loop of

*Figure 5 continued on next page*

*Figure 5 continued*

subunit D (yellow) and the P-domain and P-loop of subunit C (green). (**D**) Detail of the trimer interaction at the icosahedral threefold axis showing three E subunits (tan) and the corresponding adjacent F subunits (orange). Key residues involved in the threefold interaction, as well as W98, are indicated in stick model in the expanded view.

DOI: https://doi.org/10.7554/eLife.30822.009

the adjacent CP subunits (B and E, respectively) (*Figure 7B*), presumably reflecting the hexamer skew that shifts these subunits relative to each other (*Figure 5A*).

The SaPI1 procapsid map had the internal protrusions previously attributed to dimers of CpmB (*Dearborn et al., 2011*) (*Figure 3E*). The distal parts of the protrusions were not well ordered and appeared to correspond to a superposition of several alternate orientations of the CpmB N-terminal

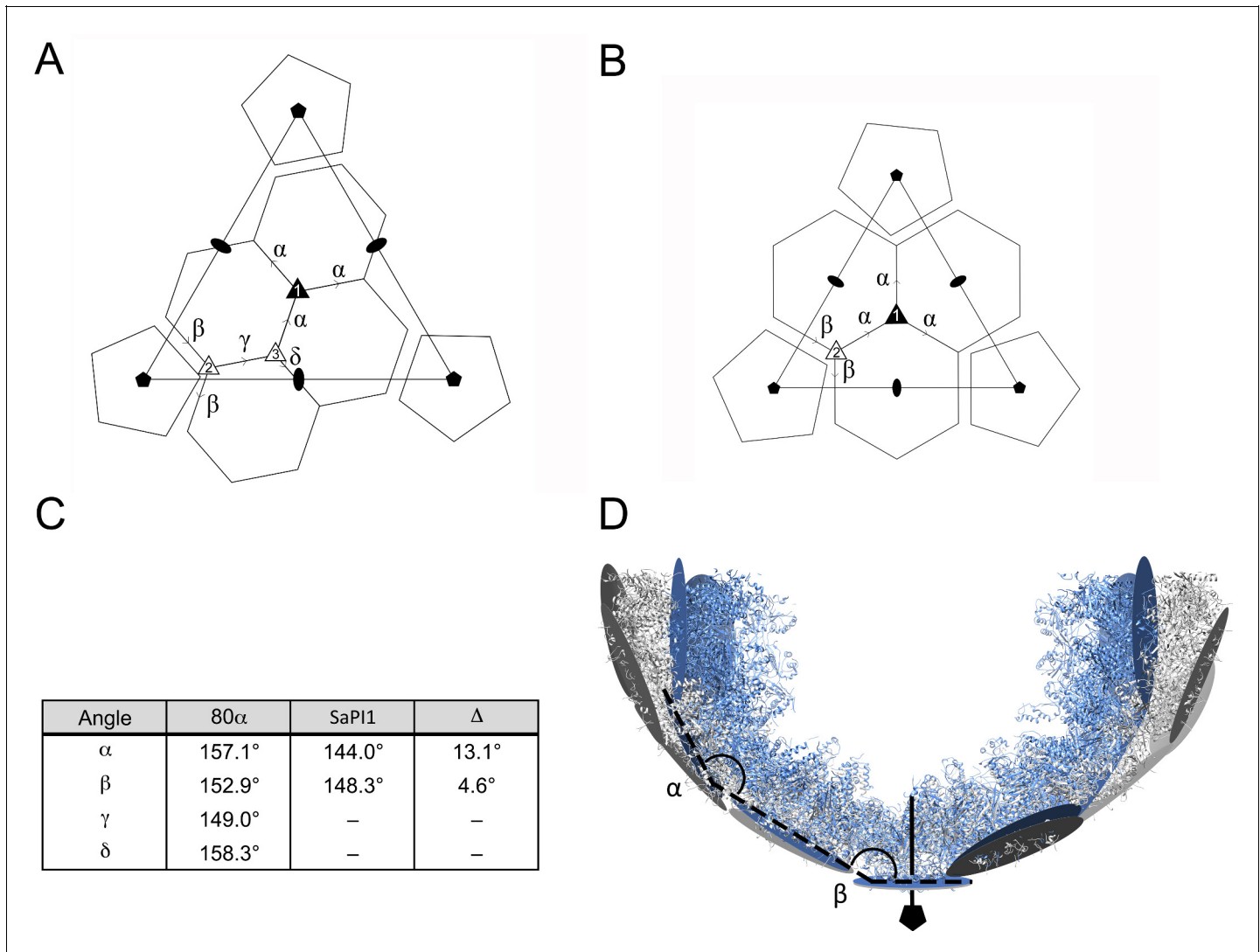

**Figure 6.** Comparison of capsomer angles in 80α and SaPI1. (**A, B**) Schematic diagrams showing the organization of hexamers and pentamers in the *T = 7* (**A**) and *T = 4* (**B**) lattices. The dihedral angles between hexamers are defined by the Greek letters α–δ. Type 1, 2 and 3 threefold axes are indicated. (**C**) Table of the calculated interior inter-capsomeric dihedral angles in 80α and SaPI1. The Δ column indicates the difference between corresponding angles. (**D**) Ribbon representation of a slab through one half of the 80α (gray) and SaPI1 (blue) shells, showing how the difference in inter-capsomer angles are propagated through the lattice. The models were aligned at one fivefold vertex (shown at bottom of diagram) and the planes representing the capsomers (gray and blue circles) and the resulting α and β angles are shown.

DOI: https://doi.org/10.7554/eLife.30822.010

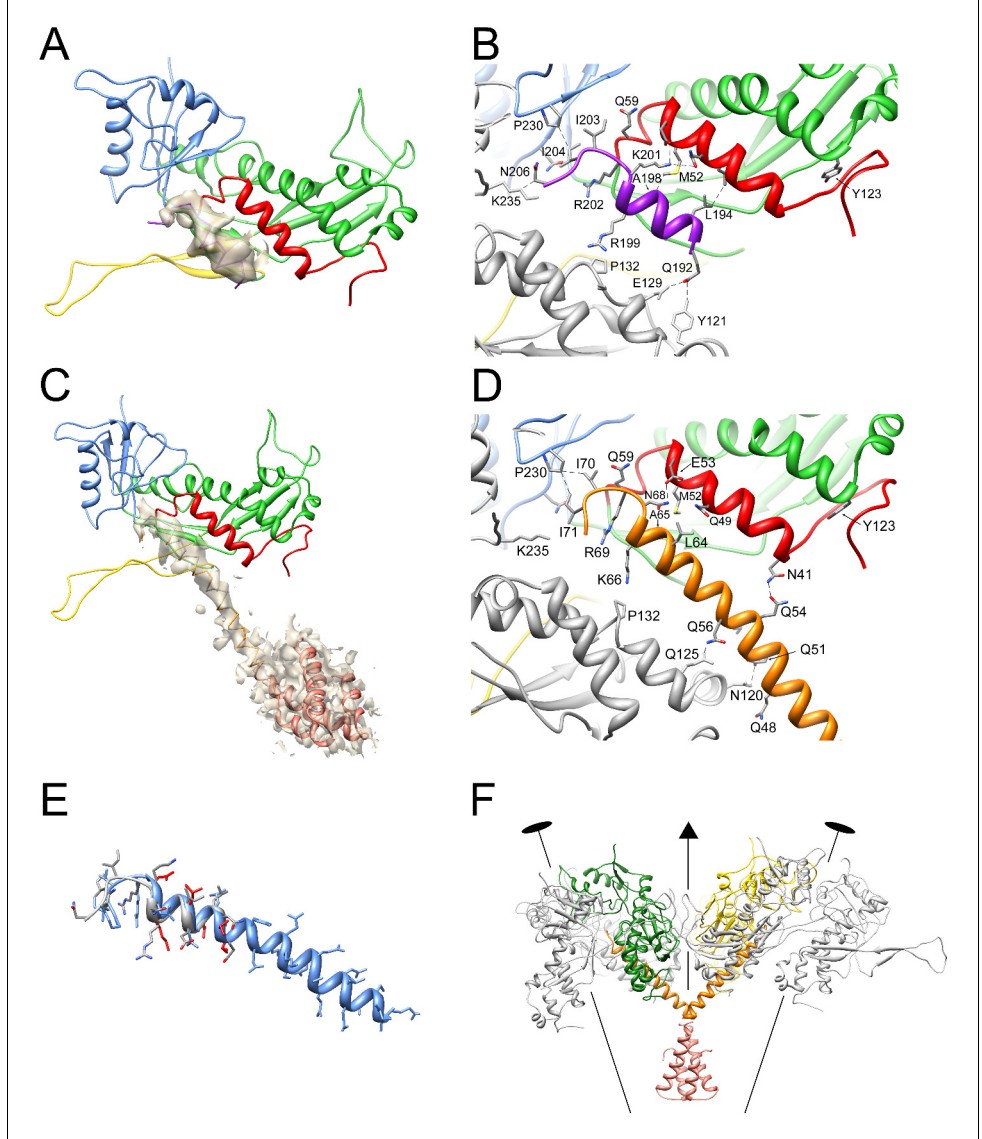

**Figure 7.** Interaction of CP with SP and CpmB. (**A**) Ribbon diagram of CP and SP from subunit C of the 80α procapsid with the density corresponding to SP superimposed, colored as in *Figure 3E*. (**B**) Detail of the SP-CP interaction, with relevant residues highlighted (stick representation) and labeled. Key interactions (<4 Å) are indicated by stippled lines. (**C**) Ribbon diagram of CP and CpmB from subunit C in the SaPI1 procapsid showing the density corresponding to CpmB, colored as in *Figure 3F*. The NMR structure of the N-terminal dimerization domain of CpmB (salmon) has been placed in the density, but could not be modeled accurately. (**D**) Detail of CpmB-CP interaction with relevant residues and distances (<4 Å) indicated. (**E**) Superposition of CpmB (blue) and SP (gray). Side chains in CpmB for residues that differ from their equivalents in SP are shown in red. (**F**) Ribbon diagram showing the interaction of one CpmB dimer (orange and salmon) with two adjacent hexameric capsomers. The C and D subunits are colored (green and yellow, respectively); other subunits are in gray. The positions of two twofold and one threefold symmetry axis are indicated.

DOI: https://doi.org/10.7554/eLife.30822.011

dimerization domain, for which we previously determined the NMR structure (*Figure 7C*). Closer to the capsid shell, the density became interpretable, and was modeled de novo as residues 41–72 of CpmB (*Figure 7D*). The C-terminal part of CpmB that includes the RIIK motif forms a helix-and-hook structure that is almost identical to that formed by SP in the 80α procapsid, and makes similar interactions with CP (*Figure 7D,E*). Contacts between CpmB and adjacent CP subunits are more extensive than in the 80α procapsid, involving several additional Gln residues in the α-helix (Q48, Q51, Q56) (*Figure 7D*). Each CpmB dimer straddles two capsomers with legs that connect with the capsid shell (*Figures 3E* and *7F*). There are two types of protrusions: one that ties together C and D

eLIFE Research article
Biophysics and Structural Biology | Microbiology and Infectious Disease

**Table 2.** List of contacts between SP and CP in the 80α procapsid (panel A, left) and between CpmB and CP in the SaPI1 procapsid (panel B, right).

Most contacts are within the same subunit (chain C), but some contacts are made with the adjacent subunit (chain B). Corresponding residues in SP and CpmB are on the same row. The assigned SP sequence starts on P191, while CpmB starts on Q48. A contact is defined as an interatomic distance of ≤4 Å. The table lists the specific residues that are involved in these contacts and enumerates the residues (# Res) and atoms (# Atoms) that contribute to these contacts. Residues that make the most contacts (≥5 interatomic pairs) are highlighted in green, while those that display ≤1 contacts are highlighted in red. 'Intermediate' contacts (2–4) are shown in yellow.

| A. 80α procapsid. contacts (≤4 Å) between SP (subunit C) and CP | | | | | B. SaPI1 procapsid. contacts (≤4 Å) between CpmB (subunit C) and CP | | | | |
| --- | --- | --- | --- | --- | --- | --- | --- | --- | --- |
| SP | CP chain C | CP chain B | Res | Atoms | CpmB | CP chain C | CP chain B | Res | Atoms |
| | | | | | Q 48 | | N120 | 1 | 5 |
| | | | | | E 49 | | | 0 | 0 |
| | | | | | E 50 | | | 0 | 0 |
| | | | | | Q 51 | | E117, N120 | 2 | 13 |
| | | | | | S 52 | | | 0 | 0 |
| | | | | | K 53 | | | 0 | 0 |
| | | | | | Q 54 | N41 | | 1 | 2 |
| | | | | | K 55 | E67 | E117, Y121 | 3 | 7 |
| | | | | | Q 56 | | Q125 | 1 | 11 |
| | | | | | Y 57 | | | 0 | 0 |
| P 191 | | | 0 | 0 | G 58 | | | 0 | 0 |
| Q 192 | | E129, Y121 | 2 | 5 | T 59 | | | 0 | 0 |
| N 193 | | | 0 | 0 | T 60 | | | 0 | 0 |
| L 194 | T45, L48, I253 | | 3 | 6 | L 61 | T45 | | 1 | 1 |
| A 195 | Y63 | | 1 | 1 | Q 62 | Y63, E64, P65 | | 3 | 17 |
| E 196 | | | 0 | 0 | N 63 | | | 0 | 0 |
| I 197 | T45, Q49, M52 | | 3 | 4 | L 64 | Q49 | | 1 | 7 |
| A 198 | M52, Y63 | | 2 | 4 | A 65 | M52, Y63 | | 2 | 5 |
| R 199 | | E50, P32 | 2 | 5 | K 66 | | P132 | 1 | 2 |
| Q 200 | | | 0 | 0 | Q 67 | | | 0 | 0 |
| K 201 | Q49, E53 | | 2 | 6 | N 68 | M52, E53 | | 2 | 5 |
| R 202 | Q59, L60, G61, K62, G247 | | 5 | 4 | R 69 | M58, Q59, L60, G61 | | 4 | 8 |
| I 203 | Q59, V232, P230 | | 3 | 7 | I 70 | Q59, P230 | | 2 | 6 |
| I 204 | L60, N194, A195, P230 | | 4 | 11 | I 71 | N194, P230 | K235 | 3 | 3 |
| K 205 | | | 0 | 0 | K 72 | | K235 | 1 | 1 |
| N 206 | | Y139, K235 | 2 | 6 | – | – | | | |

DOI: https://doi.org/10.7554/eLife.30822.012

subunits around the threefold axis, and another, weaker density, that interacts with A and B subunits, connecting hexamers and pentamers around quasi-threefold symmetry axes (*Figure 4B*).

## Mutational analysis of the SP-CP interaction

We previously made an 80α lysogen (strain ST82; see Key Resources) with *cpmA* and *cpmB* inserted into the capsid operon immediately upstream of the SP gene (ORF46) (*Damle et al., 2012*). Phage particles produced by induction of this lysogen had predominantly small capsids that were unable to package a complete 80α genome, and were thus non-infectious. We screened this lysogen for escape mutants that would evade size redirection, make infectious T = 7 particles, and form plaques. The escape mutants identified were exclusively located in the *cpmAB* insertion. Of the sixteen

mutants analyzed, five had lost the insertion (only one of these was sequenced and it retained only 174 bp at the beginning of *cpmA*), five had nonsense mutations in *cpmB*, and six were missense mutations in *cpmB*. The nonsense mutants had lost from 8 to 45 residues from the C-terminus of CpmB, demonstrating the importance of the C-terminus for CpmB function. The CpmB missense mutations included V31D, D34N, R69K, R69S, and I70T. V31 and D34 correspond to the dimer interface observed in the NMR structure of CpmB (*Dearborn et al., 2011*), suggesting that these mutants were defective in dimerization. R69 and I70 are part of the RIIK motif at the C-terminus of CpmB (*Figure 8*) and both make numerous contacts with CP in the SaPI1 procapsid structure (*Table 2*). Mutants at these sites presumably led to a failure of CpmB to interact with CP.

Based on the observed homology between the C-termini of CpmB and SP (*Figure 8*), we introduced the corresponding mutations (R202K and I203T) individually into the RIIK motif of SP (strains ST196 and ST358). Like the corresponding residues in CpmB, both residues form numerous contacts with CP in the 80α procapsid (*Figure 7*; *Table 2*). As expected, both mutations were lethal to the phage (*Table 3*). In the few plaques that formed, R202K had reverted to Arg. In order to identify second-site mutants in CP that might compensate for the SP defects, we generated R202S and R202E (strains ST278 and ST279), for which two base substitutions would be required for reversion. These mutations were also lethal (*Table 3*) and no compensatory mutations or revertants were identified. All mutations at R202 were also unable to mobilize SaPI1, demonstrating that this defect could not be compensated by wildtype CpmB (strains ST197, ST366 and ST367; *Table 3*).

For the SP I203T mutant, compensatory mutations resulting in plaque formation appeared at low frequency in both SP and CP. Pseudo-revertants of I203T included I203M, I203L and I203K, suggesting that a long, hydrophobic side chain is important at this position. I203T was also rescued by a second mutation in SP (K201I), possibly by recovering the local hydrophobicity. In the CP, compensatory mutations occurred in M52, which was repeatedly replaced by Leu and Ile, and Y123, which changed to Cys (*Figure 8*), suggesting an involvement of these residues in SP binding. Neither M52L nor Y123C introduced separately into 80α affected phage titers (strains ST384 and ST385; 32). In the 80α procapsid structure, M52 is located in helix α1 in the N-arm of CP and is not directly

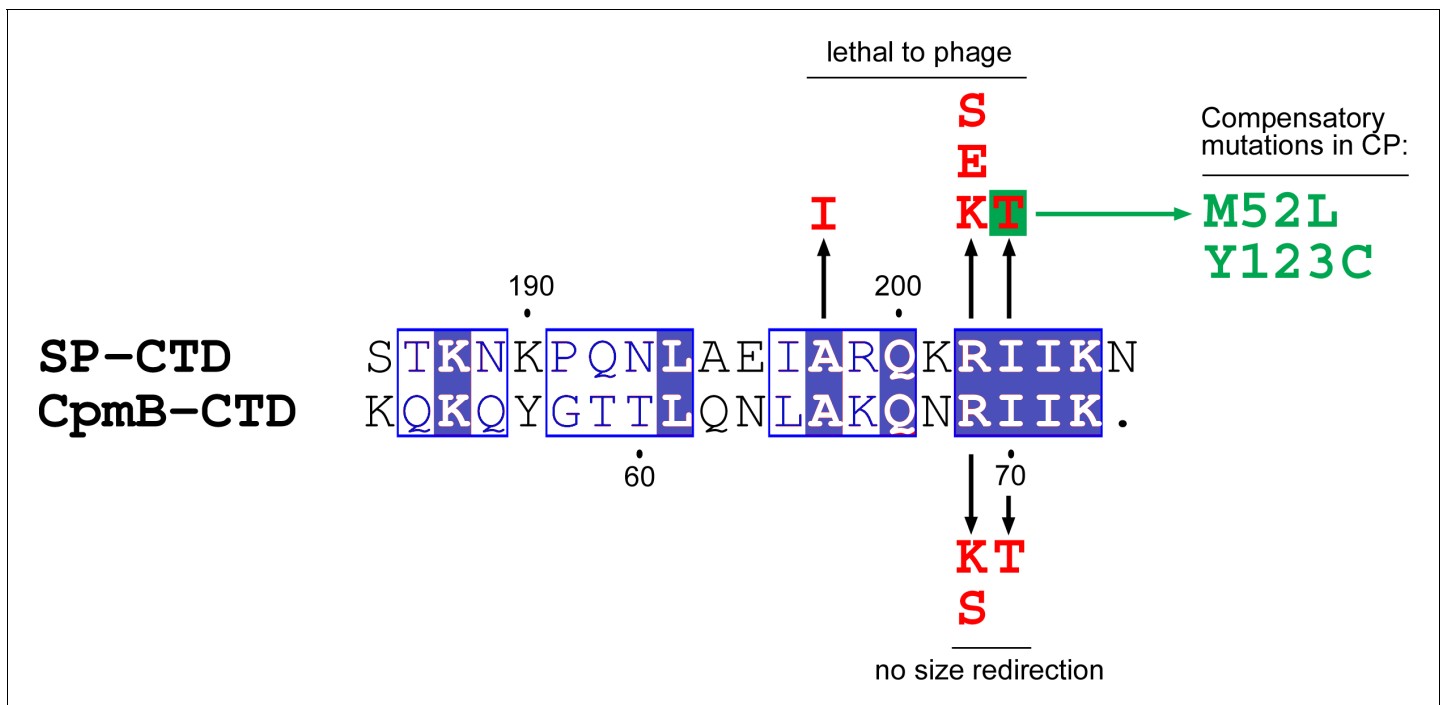

**Figure 8.** Sequence alignment of the C-terminal domains of SP and CpmB. Size redirection negative mutations in CpmB and lethal mutations in SP are indicated (red). The mutations in CP that suppressed the I203T mutant are listed (green). Alignment generated in ClustalW and rendered with ESPript 2.0.

DOI: https://doi.org/10.7554/eLife.30822.013

**Table 3.** Genetic analysis of 80α and SaPI1, listing the *S. aureus* strains and the genotypes of the 80α prophages and SaPI1 elements that they harbor.

SaPI1 also includes the *tst::tetM* insertion that confers tetracycline resistance. The corresponding phage titers (PFU/ml) and *tetM* transducing titers (TU/ml) are from filtered lysates resulting from induction with mitomycin C. Titers of mutations considered lethal or greatly impaired are shown in red; titers that are essentially wildtype are shown in green. WT = wildtype; N/D = not determined.

| Strain | Phage | SaPI1 | Phage titer | SaPI1 titer |
| --- | --- | --- | --- | --- |
| RN10616 | WT | – | 8.70E + 10 | – |
| RN10628 | WT | WT | 5.10E + 08 | 1.33E + 08 |
| | | | | |
| ST196 | SP R202K | – | <10 | – |
| ST197 | SP R202K | WT | N/D | <10 |
| ST278 | SP R202S | – | <10 | – |
| ST366 | SP R202S | WT | <10 | 1.67E + 01 |
| ST279 | SP R202E | – | <10 | – |
| ST367 | SP R202E | WT | N/D | <10 |
| ST358 | SP I203T | – | 1.00E + 02 | – |
| ST368 | SP I203T | WT | 1.71E + 02 | 6.67E + 06 |
| ST469 | SP I204L | – | 5.33E + 10 | – |
| ST417 | SP A198I | – | 1.00E + 01 | – |
| ST454 | SP A198I | WT | <10 | 4.18E + 06 |
| | | | | |
| ST384 | CP M52L | – | 1.30E + 10 | – |
| ST415 | CP M52Q | – | 1.00E + 01 | – |
| ST481 | CP M52Q | WT | <10 | 3.00E + 01 |
| ST385 | CP Y123C | – | 1.47E + 10 | – |
| | | | | |
| ST466 | SP::CpmB$_{CTD}$ | – | 2.70E + 09 | – |
| ST467 | SP::CpmB$_{CTD}$ | WT | 9.00E + 06 | 9.25E + 05 |
| ST468 | SP::CpmB$_{CTD}$ | CpmB::SP$_{CTD}$ | 6.80E + 06 | 9.00E + 05 |
| ST458 | WT | CpmB::SP$_{CTD}$ | 1.80E + 08 | 9.60E + 08 |
| ST465 | SP R202E | CpmB::SP$_{CTD}$ | <10 | 1.40E + 02 |

DOI: https://doi.org/10.7554/eLife.30822.014

involved in interactions with SP I203T or other residues in the RIIK motif. Instead, it interacts with I197 and A198 in the SP α-helix (*Figure 7B*). The M52L mutation might therefore strengthen these hydrophobic interactions. Consistent with this, an M52Q mutant was lethal to the phage (ST415) and was unable to mobilize SaPI1 (ST481; *Table 3*). Mutation of A198 to Ile was also lethal to the phage, presumably because there is insufficient space for the bulky side chain (ST417; *Table 3*). The other site of compensatory mutations, Y123, is not involved in SP interactions, but instead interacts with residues in the N-arm of CP (*Figure 7B*), suggesting a more indirect effect on SP binding.

Surprisingly, both 80α SP mutants I203T and A198I (strains ST368 and ST454, respectively) mobilized SaPI1 with near-wildtype (RN10628) transducing titers (*Table 3*), producing capsids that were predominantly small (99% for ST368, n = 119; 97% for ST454, n = 176), although for ST368 (I203T), most of the capsids (86%) were procapsids rather than full virions (*Figure 9A,B*). Apparently, CpmB is able to compensate for certain lethal defects in SP, presumably by providing the required scaffolding interaction with CP. In light of this, it is perhaps surprising that SaPI1 could not compensate for mutations at R202. Presumably, the R202 defect extends beyond the scaffolding function of SP that

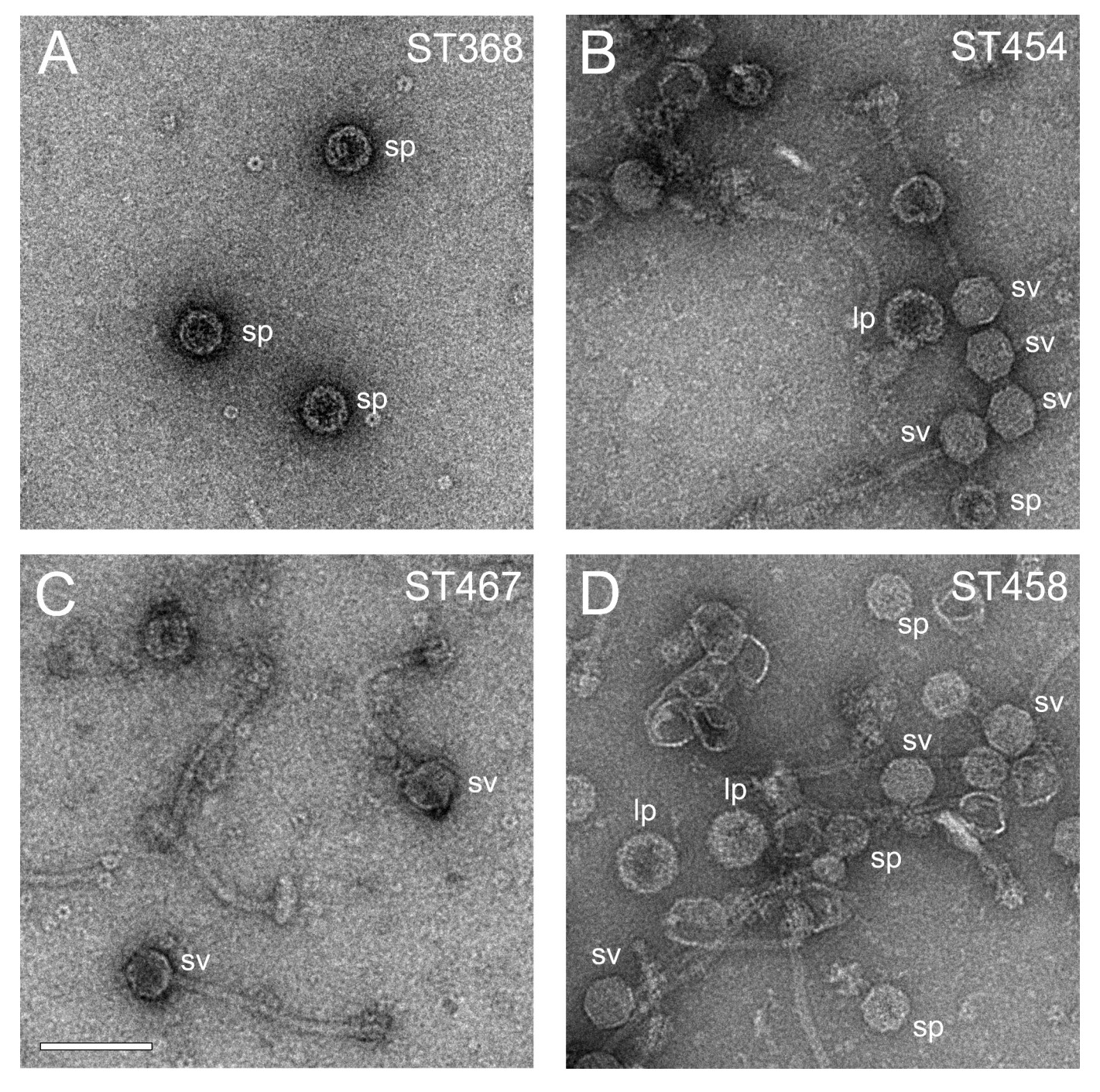

**Figure 9.** Electron micrographs of negatively stained particles from mutant lysates. (**A**) Strain ST368 (80α SP I203T + wildtype SaPI1); (**B**) strain ST454 (80α SP A198I + wildtype SaPI1; (**C**) strain ST467 (80α SP-CpmB$_{CTD}$ + wildtype SaPI1); (**D**) strain ST458 (80α wildtype + SaPI1 CpmB-SP$_{CTD}$). Examples of small procapsids (sp), small virions (sv), and large procapsids (lp) are indicated on the images. Scale bar denotes 100 nm.
DOI: https://doi.org/10.7554/eLife.30822.015

can be compensated by CpmB. For example, R202 may be involved in portal incorporation or interaction with CpmA.

The structural and mutational data suggested that CpmB and SP bind to CP using essentially the same interactions, mediated via the C-terminal 16 residues that include the conserved RIIK motif and

the preceding α-helix (*Figure 7*). We hypothesized that the small sequence differences between the two proteins (*Figure 8*) would allow CpmB to outcompete SP for binding to CP. This would allow SaPI1 to promote small capsid formation even when SP is present, as is the case during a normal SaPI1 mobilization event.

We tested this hypothesis by swapping the C-termini of SP (residues 186–206) and CpmB (residues 53–72), generating phage 80α SP-CpmB$_{CTD}$ and SaPI1 CpmB-SP$_{CTD}$. The strains containing the chimeric 80α and SaPI1 are listed in *Table 3*. The 80α SP-CpmB$_{CTD}$ phage (strain ST466) had a slightly reduced phage titer compared to the wildtype phage (RN10616), and mobilized wildtype SaPI1 with a slightly reduced transducing titer (strain ST467) compared to the wildtype (RN10628; *Table 3*), showing that the C-terminal domain of CpmB functions almost as well as the native SP for interaction with CP. In the presence of SaPI1, the majority (96%, n = 24) of the capsids produced were small (*Figure 9C*), showing that the ability of CpmB to redirect the assembly pathway was not impaired in the presence of the chimeric SP. The same result was found when 80α SP-CpmB$_{CTD}$ was used to mobilize the chimeric SaPI1 CpmB-SP$_{CTD}$ (strain ST468; 100% small, n = 60). When SaPI1 CpmB-SP$_{CTD}$ was mobilized by wildtype phage (strain ST458), the transducing titer was normal (*Table 3*). This was expected, since SaPI1 does not require a functional CpmB to transduce normally (*Ubeda et al., 2008*). The chimeric CpmB-SP$_{CTD}$ produced fewer small capsids (90%, n = 380) (*Figure 9D*) than the other chimeras, suggesting an impaired ability of the chimeric CpmB-SP$_{CTD}$ to compete with the wildtype SP. (The difference was statistically significant (p<0.05) compared to ST468.) The phage titer in this strain was at wildtype levels, presumably due to the presence of the wildtype SP (*Table 3*). The chimeric CpmB-SP$_{CTD}$ was unable to compensate for the R202E defective SP (strain ST465), consistent with an additional role for this residue.

## Discussion

We have solved the near-atomic resolution structures of the *S. aureus* bacteriophage 80α procapsid and the smaller procapsid produced in the presence of the SaPI1 genomic element. These structures show, for the first time, the detailed interactions between a bacteriophage scaffolding protein and the capsid, and provide unique insights into the assembly and size determination process of dsDNA bacteriophages and macromolecular complexes in general.

Internal scaffolding proteins are used by essentially all members of the Caudovirales. In some cases, the SP is part of another protein, such as the HK97-like phages, which have a functionally equivalent domain fused to their CP (*Conway et al., 1995*). In bacteriophage P2 and its relatives, the SP is part of the protease (*Chang et al., 2010*). An exception is the encapsulins, phage-like nanocompartments with *T* = 3 symmetry that do not undergo maturation, but remain in a procapsid state (*McHugh et al., 2014*). Generally, SPs are thought to curtail the inherent conformational flexibility that allows viral capsid proteins to assemble in many different ways (including aberrant capsid structures or 'crapsids') (*Dokland, 1999*; *Dokland, 2000*).

Structures of scaffolding proteins are scarce, but generally have a high α-helical content, a propensity for coiled-coil formation, and high flexibility (*Dokland, 1999*; *Prevelige and Fane, 2012*). CpmB resembles the gp9 scaffolding protein of bacteriophage φ29, which also forms a dimeric, α-helical bundle (*Morais et al., 2003*; *Dearborn et al., 2011*). The bacteriophage P22 SP features an RKLK sequence in a helix-loop-helix motif that was genetically determined to interact with the P22 CP (*Cortines et al., 2014*). However, the exact interactions between these proteins and their capsids could not be resolved in the cryo-EM structures due to insufficient resolution (*Morais et al., 2003*; *Chen et al., 2011*).

In the 80α procapsid, the C-terminal 21 residues of SP form a 'helix-and-hook' motif that interacts extensively with the CP, especially with the N-arm (*Figure 7*). The rest of the SP was not resolved in the high-resolution map, suggesting that it is disordered and/or not organized with icosahedral symmetry. This is consistent with the previous low-resolution structure, which showed a diffuse core separated from the shell by a 25 Å gap (*Spilman et al., 2011*). Most likely, this gap is traversed by a flexible linker domain that connects this core to the helix-and-hook motif. In the absence of SP, no capsids are formed and CP levels are greatly repressed, suggesting that SP acts as a chaperone for correct folding and to prevent aggregation of CP (*Spilman et al., 2012*). Minor changes in the SP that disrupted CP interactions (e.g. R202K, I203T) completely abolished its function (*Table 3*).

In the SaPI1 procapsid, the SP is replaced by CpmB, which forms a helix-and-hook motif that is very similar to SP, and interacts with CP via similar interactions (*Figure 7*). By using the same interactions as SP, the helper phage is prevented from mutating its CP to escape the size redirection, since this would preclude SP binding as well, leading to a lethal phenotype. Consequently, no CpmB-resistant escape mutants in CP were identified in our genetic screens. If the two proteins compete directly for a binding site on CP, the effectiveness of size redirection would depend on CpmB having a higher affinity for CP than SP. Indeed, CpmB makes additional contacts with CP that were not observed with SP; however, these contacts cannot be compared directly, since the corresponding SP residues were not resolved in the 80α map. Consistent with a competition model, SaPI1 with a chimeric CpmB carrying the C-terminus of SP (CpmB-SP$_{CTD}$) produced more large capsids than wild-type CpmB (Strain ST458; *Figure 9D*).

Once bound to CP, how does CpmB induce the formation of small capsids? Comparison of the 80α and SaPI1 procapsid structures shows that the hexameric capsomers are very similar, in spite of the different environments that they occupy in the two capsids (*Figure 4D*). The pentamers have similar environments in both capsids and were expected to be similar. Clearly, CpmB does not affect the structure of the capsomers themselves. Instead, large and small procapsids differ primarily in the way that the hexamers and pentamers interact with each other around the icosahedral and quasi-threefold axes of symmetry (*Figure 6*). The small capsids have a more acute dihedral angle between capsomers, especially around the icosahedral threefold axis (*Figure 6*), where we previously showed that CpmB dimer occupancy was higher (*Dearborn et al., 2011*). CpmB dimers constrain these angles by straddling adjacent capsomers, whereas the longer, more flexible N-terminal domain of SP apparently does not constrain the capsomers in the same way. In a flat plane, hexamers can be incorporated *ad infinitum*, but on a curved surface, pentamers will have to be incorporated at specific points in order to close the lattice (*Figure 10*). It should be noted that the observed hexamer skew is also a necessary consequence of placing the hexamers on a curved surface. In the mature shell, the icosahedral faces are almost flat, and capsomers are symmetrical (*Spilman et al., 2011*). In

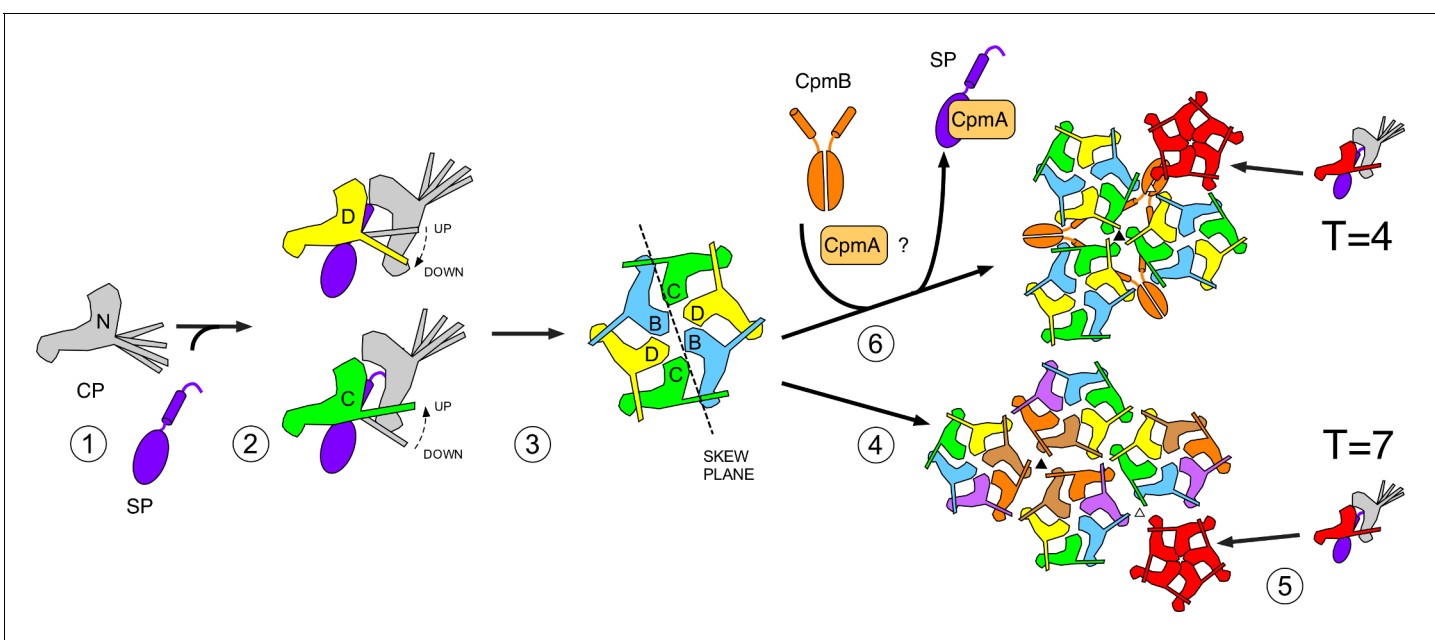

**Figure 10.** Schematic model for scaffolding competition-mediated capsid size redirection. In this scheme, SP is needed for the correct folding of CP (1), which is only stable as an oligomer. The E-loop conformation is 'up' or 'down' depending on its binding to a neighboring CP subunit (2). SP might be involved in this conformational switching. Subunits are assembled into hexamers (and possibly other oligomers), which are independent of the presence or absence of CpmB (3). The curvature imposed on the nascent shell by the trivalent interaction of hexamers (4) forces the incorporation of pentamers into the lattice at specific points (5), leading to a T = 7 shell. The presence of CpmB dimers imposes a sharper angle between hexamers (6), leading to the more frequent incorporation of pentamers and the formation of a the smaller, T = 4 shell. CpmA is thought to facilitate the removal of SP to provide access for CpmB (6).
DOI: https://doi.org/10.7554/eLife.30822.016

the absence of CpmB, the less curved shell allows the incorporation of additional hexamers before requiring a pentamer, resulting in the $T = 7$ shell (*Figures 6* and *10*). In the presence of CpmB, the more curved trimer of hexamers would need to incorporate pentamers more frequently, resulting in the smaller, $T = 4$ shell (*Figures 6* and *10*).

We previously showed small capsid formation requires CpmA as well as CpmB (*Dearborn et al., 2011*; *Damle et al., 2012*; *Spilman et al., 2012*). The role of CpmA in the size redirection process is still unclear. Co-expression of CpmA with CP and SP in the absence of CpmB led to a failure to assemble capsids (*Spilman et al., 2012*), and a SaPI1 Δ*cpmB* mutant produced a large fraction of aberrant shells (*Dearborn et al., 2011*). One possibility is that CpmA is required to remove SP from its binding site on CP and thereby provide access for CpmB (*Figure 10*). Perhaps CpmA is needed to disrupt an internal core that would otherwise prevent the formation of a smaller shell.

SP is required for the formation of viable transducing particles even in the presence of a complete SaPI1 element (*Spilman et al., 2012*), indicating that there are some functions of SP that cannot be provided by CpmB. Consequently, although CpmB could compensate for certain lethal SP mutants (I203T and A198I), it was unable to compensate for mutations at R202. SP may be required early in the assembly process to ensure the correct folding of CP, or to control the conformational switching during formation of hexamers and pentamers (*Figure 10*). It is also likely that SP is required for incorporation of the portal, by analogy with bacteriophage φ29 (*Fu et al., 2007*). Although well-formed capsids can be assembled without portals, it is likely that portals are involved in the initiation of shell assembly during phage production in vivo. Some of these functions presumably reside in the N-terminal part of the protein that was not observed in the 80α procapsid map.

Restricting the size of the capsids is only one mechanism that the SaPIs use to suppress phage multiplication during molecular piracy (*Christie and Dokland, 2012*; *Novick and Ram, 2016*). Indeed, SaPIs are perfectly viable even if they are prevented from making small capsids, and interference with phage replication is similar. Nevertheless, *cpmAB* genes are present and highly conserved between a large number of SaPIs (*Novick et al., 2010*). Recently, a different mechanism of capsid size redirection that is not dependent on CpmAB was discovered in a distinct group of SaPIs that are mobilized by *cos* site phages (*Carpena et al., 2016*). This example of convergent evolution suggests that size redirection is an important aspect of the long-term survival and establishment of SaPIs in the bacterial population. The SaPIs, in turn, have a profound effect on the emergence of bacterial virulence and antibiotic resistance. In either case, the respective 'helper' phages are themselves helpless in the face of these molecular pirates.

# Materials and methods

## Key resources

| Reagent type (species) or resource | Designation | Source or reference | Identifiers | Additional information |
|---|---|---|---|---|
| strain, strain background (*Staphylococcus aureus*) | RN4220 | Network onantimicrobial resistance in *Staphylococcus aureus* | NARSA: NRS144 | Obtained from Richard Novick |
| strain, strain background (*S. aureus*) | RN10616 | PMID: 19347993 | | RN4220(80α); Obtained from Richard Novick |
| strain, strain background (*S. aureus*) | RN10628 | PMID: 19347993 | | RN4220(80α) SaPI1; Obtained from Richard Novick |
| strain, strain background (*S. aureus*) | ST24 | PMID: 19347993 | | RN4220 (80α Δ*terS*) |
| strain, strain background (*S. aureus*) | ST65 | PMID: 21821042 | | RN4220 (80α Δ*orf44*) SaPI1 |
| strain, strain background (*S. aureus*) | ST82 | PMID: 22709958 | | RN4220 (80α::SaPI1*cpmAB*) |
| strain, strain background (*S. aureus*) | ST91 | PMID: 22980502 | | RN4220 (80α Δ*SP*) |

*Continued on next page*

Continued

| Reagent type (species) or resource | Designation | Source or reference | Identifiers | Additional information |
|---|---|---|---|---|
| strain, strain background (*S. aureus*) | ST100 | PMID: 21821042 | | RN4220 (80α) (SaPI1Δ*cpmB*) |
| strain, strain background (*S. aureus*) | ST196 | this paper | | RN4220 (80α SP::R202K) |
| strain, strain background (*S. aureus*) | ST197 | this paper | | RN4220 (80α SP::R202K) SaPI1 |
| strain, strain background (*S. aureus*) | ST248 | this paper | | RN4220 (80α Δ*CP*) |
| strain, strain background (*S. aureus*) | ST278 | this paper | | RN4220 (80α SP::R202S) |
| strain, strain background (*S. aureus*) | ST279 | this paper | | RN4220 (80α SP::R202E) |
| strain, strain background (*S. aureus*) | ST358 | this paper | | RN4220 (80α SP::I203T) |
| strain, strain background (*S. aureus*) | ST366 | this paper | | RN4220 (80α SP::R202S) SaPI1 |
| strain, strain background (*S. aureus*) | ST367 | this paper | | RN4220 (80α S::R202E) SaPI1 |
| strain, strain background (*S. aureus*) | ST368 | this paper | | RN4220 (80α SP::I203T) SaPI1 |
| strain, strain background (*S. aureus*) | ST384 | this paper | | RN4220 (80α CP::M52L) |
| strain, strain background (*S. aureus*) | ST385 | this paper | | RN4220 (80α CP::Y123C) |
| strain, strain background (*S. aureus*) | ST415 | this paper | | RN4220 (80α CP::M52Q) |
| strain, strain background (*S. aureus*) | ST417 | this paper | | RN4220 (80α SP::A198I) |
| strain, strain background (*S. aureus*) | ST454 | this paper | | RN4220 (80α SP::A198I) SaPI1 |
| strain, strain background (*S. aureus*) | ST458 | this paper | | RN4220 (80α) SaPI1 CpmB::SP$_{CTD}$ |
| strain, strain background (*S. aureus*) | ST465 | this paper | | RN4220 (80α SP::R202E) SaPI1 CpmB::SP$_{CTD}$ |
| strain, strain background (S. aureus) | ST466 | this paper | | RN4220 (80α SP::CpmB$_{CTD}$) |
| strain, strain background (*S. aureus*) | ST467 | this paper | | RN4220 (80α SP::CpmB$_{CTD}$) SaPI1 |
| strain, strain background (*S. aureus*) | ST468 | this paper | | RN4220 (80α SP::CpmB$_{CTD}$) SaPI1 CpmB::SP$_{CTD}$ |
| strain, strain background (*S. aureus*) | ST481 | this paper | | RN4220 (80α CP::M52Q) SaPI1 |
| commercial assay or kit | In-Fusion HD cloning kit | Clontech | Clontech: Cat # 639646 | http://www.clontech.com/US/ Products/Cloning_and_Competent_ Cells/Cloning_Kits/Older_Cloning _Kits?sitex=10020:22372:US |
| strain, strain background (*Escherichia coli*) | Stellar | Clontech | Clontech: Cat # 636763 | Competent E. coli HST08 strain |
| recombinant DNA reagent | pMAD (plasmid) | PMID: 15528558 | | Obtained from Richard Novick; used to generate all *S. aureus* strains listed above |

*Continued on next page*

*Continued*

| Reagent type (species) or resource | Designation | Source or reference | Identifiers | Additional information |
|---|---|---|---|---|
| recombinant DNA reagent | pEW3 (plasmid) | this paper | | Made in pMAD by In-Fusion cloning |
| recombinant DNA reagent | pEW14 (plasmid) | this paper | | Made in pMAD by In-Fusion cloning |
| recombinant DNA reagent | pEW19 (plasmid) | this paper | | Made in pMAD by In-Fusion cloning |
| recombinant DNA reagent | pEW20 (plasmid) | this paper | | Made in pMAD by In-Fusion cloning |
| recombinant DNA reagent | pLAK1 (plasmid) | this paper | | Made in pMAD by In-Fusion cloning |
| recombinant DNA reagent | pLKP2 (plasmid) | this paper | | Made in pMAD by In-Fusion cloning |
| recombinant DNA reagent | pLKP3 (plasmid) | this paper | | Made in pMAD by In-Fusion cloning |
| recombinant DNA reagent | pLKP14 (plasmid) | this paper | | Made in pMAD by In-Fusion cloning |
| recombinant DNA reagent | pLKP15 (plasmid) | this paper | | Made in pMAD by In-Fusion cloning |
| recombinant DNA reagent | pLKP31 (plasmid) | this paper | | Made in pMAD by In-Fusion cloning |
| recombinant DNA reagent | pLKP32 (plasmid) | this paper | | Made in pMAD by In-Fusion cloning |
| software, algorithm | EMAN2 (computer program suite) | PMID: 16859925 | | Downloaded from http://blake.bcm.edu/emanwiki/EMAN2 |
| software, algorithm | jspr (computer program suite) | PMID: 24357374 | | Downloaded from http://jiang.bio.purdue.edu/jspr |
| software, algorithm | BSOFT (computer program suite) | PMID:17011211 | | Downloaded from https://lsbr.niams.nih.gov/bsoft/bsoft_download.html |

## Bacterial culture

Bacterial strains used in this work are listed under key resources. *S. aureus* cultures were grown in Tryptic Soy Broth (TSB) or on Tryptic Soy Agar (TSA) plates (1.8% agarose) at 32°C. When necessary, the media were supplemented with erythromycin (5 µg/ml) or tetracycline (5 µg/ml). *Escherichia coli* strains were grown in lysogeny broth (LB) or on LB agar at 37°C and supplemented with ampicillin (100 µg/ml) when needed.

## DNA manipulations

All plasmids in this study were constructed via Gibson assembly using the Clontech In-Fusion HD kit (Takara Bio USA, Mountain View, CA). Primers used are listed in *Supplementary file 1*. Gel and PCR purifications were performed using the Nucleospin Gel and PCR Clean-up Kit (Macherey-Nagel, Bethlehem, PA) in accordance with the manufacturer's instructions. Plasmid DNA preparations were performed using the QIAprep Spin Miniprep Kit (Qiagen, Valencia, CA) in accordance with the manufacturer's instructions. All plasmids were verified by Sanger sequencing (MWG Operon, Huntsville, AL).

## Preparation of competent cells and transformation of *S. aureus*

Electro-competent *S. aureus* cells were prepared as in *Lee, 1995* and stored at −80°C in 60 µl aliquots. After thawing aliquots on ice, 0.3–0.6 µg plasmid DNA was added. After 30 min on ice, mixture was transferred to a 0.1 cm Gene Pulser cuvette and electroporated on the STA setting (2.50 kV for one pulse, 2.5 ms) of a MicroPulser electroporation apparatus (Bio-Rad, Hercules, CA).

Immediately after the pulse, cells were transferred to a 2 ml microcentrifuge tube containing 1 ml brain heart infusion broth (BHI). Cells were recovered by shaking for 2 hr at 200 rpm, 30°C, and then plated on TSA plates supplemented with appropriate antibiotics and/or X-gal, as needed.

## Allelic exchange

Allelic exchange was done using the pMAD shuttle vector, as previously described (*Arnaud et al., 2004*; *Poliakov et al., 2008*). The desired mutations were introduced into pMAD using the Clontech In-Fusion HD cloning kit. Flanking DNA fragments containing the mutation of interest and homologous overlapping sequences were amplified via PCR (*Supplementary file 1*). The purified PCR products were combined with the In-Fusion enzyme and NcoI-digested pMAD. Plasmids were then transformed into electrocompetent *S. aureus* strains with the appropriate deletions (ST91, ST100 or ST248; see Key Resources). The cells were plated on TSA with 5 µg/ml erythromycin and 200 µg/ml X-Gal at 42°C. Resultant blue colonies were selected and grown at 30°C in TSB without erythromycin, followed by plating at 42°C to cure the cells of the plasmid. White colonies were screened by PCR and sequencing to confirm the desired mutation.

## Phage propagation and titering

Lysogenic *S. aureus* strains were grown at 32°C in 25:1 CY media with β-glycerophosphate (*Novick, 1991*). At $A_{600} \approx 0.6$ OD, the cells were diluted 1:10 into a 1:1 mixture of CY + β-glycerophosphate and *S. aureus* phage buffer (50 mM Tris-HCl pH 7.8, 100 mM NaCl, 1 mM $MgSO_4$ and 4 mM $CaCl_2$). Lysis was induced via UV irradiation (20 s under UV in a Nuaire biological safety cabinet) or with mitomycin C (2 µg/ml), followed by shaking at 100 rpm at 32°C until lysis occurred (around 3 hr). Lysates were sterile-filtered and kept at 4°C. Serial dilutions of the lysate in *S. aureus* phage buffer were plated with 100 µl overnight culture of RN4220 on *S. aureus* phage agar (*Novick, 1991*) using soft agar overlay supplemented with 0.5 mM $CaCl_2$. All plaque assays were performed in triplicate. Plaque purification was performed with plaques cored from a plate using a sterile Pasteur pipette, resuspended in 1 ml of phage buffer, serially diluted and plated. This process was repeated twice. The final plaque was resuspended in 100 µl water and used as template for PCR amplification and for sequencing.

## Transduction assays

To make double lysogens containing mutant 80α and wildtype SaPI1, 100 µl of the mutant 80α lysogen were combined 1:1 with serial dilutions of a filtered RN10628 lysate, and incubated for 15 min at 22°C. The entire volume (200 µl) was spread onto GL agar plates supplemented with 0.17 mM Na citrate and 5 µg/ml tetracycline (*Novick, 1991*) and incubated for 48 hr at 30°C. The resulting transductants were checked by PCR and sequencing for the presence of both 80α and SaPI1. To check for mobilization of mutant and wildtype SaPI1, the double lysogens were induced with mitomycin C. The resulting lysate was filtered, combined with RN4220 and plated on GL agar with 5 µg/ml tetracycline. Resulting colonies were counted to yield a transducing titer. The values reported are the average of at least three independent determinations.

## Preparation of cryo-EM samples

80α procapsids were produced by mitomycin induction of the *S. aureus* 80α lysogen ST24 (see Key Resources), which has a deletion of the small terminase (*terS*) gene, as previously described (*Spilman et al., 2011*). After lysis, cell debris was removed by centrifugation at 7,000 g for 20 min. The procapsids were collected by precipitation with 10% PEG 8,000 (w/v) and 0.5 M NaCl, followed by centrifugation at 7,000 g for 20 min. The pellet was resuspended in phage buffer, made up to 1.42 g/cm$^3$ CsCl, and centrifuged for 20 hr at 70,000 rpm in a Beckman NVT90 rotor at 15°C. The procapsid band was collected with a needle and dialyzed into phage dialysis buffer (25 mM Tris-HCl pH 7.8, 50 mM NaCl, 1 mM $MgSO_4$ and 4 mM $CaCl_2$). Separation of procapsids from phage tails was achieved by centrifugation through a 10–40% (w/v) sucrose gradient in phage dialysis buffer for 2 hr at 30,000 rpm in a Beckman SW41 rotor at 4°C. The procapsid-containing fractions were identified by SDS-PAGE, pooled and pelleted for 1 hr at 50,000 rpm in a Beckman Type 70Ti rotor at 4°C. The pellet was resuspended in phage dialysis buffer.

SaPI1 procapsids were produced similarly by induction of strain ST65, which includes SaPI1 *tst:: tetM* and 80α with a deletion of *orf44*, which encodes the minor capsid protein gp44 (see Key Resources). While gp44 is essential to 80α, deletion of *orf44* has no effect on SaPI1 viability (*Dearborn et al., 2011*). ST65 produces procapsids as well as virions, which are readily separated by CsCl density gradient centrifugation. Purification was as described for 80α procapsids. The resulting sample was contaminated with about 50% empty, expanded capsids, presumably resulting from virions that had lost their DNA. However, these capsids were easily distinguishable in the micrographs.

## Electron microscopy

For negative stain EM, crude lysates were clarified by centrifugation at 5,000 g for 20 min, followed by pelleting for 1 hr at 50,000 rpm in a Beckman 70Ti rotor. The pellets were resuspended in phage dialysis buffer at approximately 0.1 mg/ml concentration. Samples were applied to glow-discharged continuous carbon grids, washed 2x with $H_2O$ and stained with 1% uranyl acetate before imaging in an FEI Tecnai F20 electron microscope equipped with a Gatan Ultrascan 4000 CCD camera.

Samples for cryo-EM, at a protein concentration of 1 mg/ml in phage dialysis buffer, were placed on glow-discharged (40 mA, 15 s) 200 mesh nickel Quantifoil R2/1 grids and vitrified in liquid ethane using a Vitrobot Mark IV with 5 s blot time and blot pressure of 5, at 80% humidity. Grids were checked for ice quality and concentration in an FEI Tecnai F20 microscope equipped with a Gatan 626 side entry holder, before being shipped to the Biological Science Imaging Resource (BSIR) at Florida State University in a dry cryogenic shipper. At BSIR, the grids were imaged with an FEI Titan Krios electron microscope equipped with a Direct Electron DE-20 detector operated in integrating mode, at a nominal magnification of 29,000x, resulting in 1.21 Å/pixel sampling. Data were collected at 32 frames per second and a total dose of $\approx 30$ e−/Å$^2$. Frames were aligned and integrated using the DE_process_frames script (v. 2.8.1) from Direct Electron, using a quanta value of 3.

## Three-dimensional reconstruction and model building

Particles were picked semi-automatically using EMAN2 (*Tang et al., 2007*). Contrast transfer function parameters were determined and particle stacks were phase flipped in EMAN2. No amplitude correction was made. Icosahedral reconstruction was done using the program *jspr*, mainly following the procedures outlined in *Guo and Jiang (2014)*. The particles were divided into two half-sets at the outset, which were processed independently. Starting models for each set were generated using the random orientation generation procedure in *jspr* (*Guo and Jiang, 2014*). The final data sets included 10,527 particles for 80α and 14,087 particles for SaPI1. The final maps were generated from the re-combined half datasets and sharpened by application of an empirical inverse B factor using *bfilter* from the BSOFT suite (*Heymann and Belnap, 2007*). Local resolution in both maps was calculated using ResMap (*Kucukelbir et al., 2014*).

The previously generated pseudoatomic model for the 80α (*Spilman et al., 2011*) was used as a starting point for modeling of CP. SP was modeled de novo from the density. The 80α model was initially built in O and refined using Phenix with default parameters. The SaPI1 procapsid model was built in Coot with the 80α model as a starting point, and refined by iterative real-space refinement in Coot and by reciprocal space refinement in REFMAC5 (*Brown et al., 2015*). This model was placed back into the 80α procapsid map and refined with Coot and REFMAC5. Model geometry was monitored with MolProbity (*Chen et al., 2010*). To uncover any overfitting issues, the atomic coordinates of the 80α and SaPI1 models were randomly displaced by up to 0.5 Å and refined against the reconstruction from one of the two initial half sets of particles (FSCwork). These re-refined models were then compared to the reconstructions of both initial sets of particles (FSCwork and FSCtest; *Figure 3—figure supplement 2*), according to *Brown et al., 2015*. UCSF Chimera was used for manipulating and aligning maps and models, calculating RMSD values, calculating and displaying molecular contacts, calculating capsomer planes, and generating figures (*Pettersen et al., 2004*). Intercapsomeric angles were determined from calculated plane normal vectors, and model surface areas were calculated using *areaimol* in the CCP4 suite of programs (*Winn et al., 2011*).

The electron density and atomic models were submitted to EMDB with accession numbers EMDB-7030, PDB ID: 6B0X for 80αand EMDB-7035, PDB ID: 6B23 for SaPI1.

## Acknowledgements

The authors acknowledge Ms. Cindy Rodenburg and Dr. Rosie Hill for assistance with preparative aspects of this work. Cryo-EM sample preparation and screening was done at the UAB Cryo-EM core facility. High-resolution cryo-EM data was collected at the Biological Science Imaging Resource (BSIR) at Florida State University, which was supported by NIH grants S10 OD018142 and S10 RR025080 (Kenneth Taylor, PI). ADD was supported by the Intramural Research Program of the National Institute of Arthritis and Musculoskeletal and Skin Diseases of The National Institutes of Health. This project was funded by The National Institutes of Health award R01 AI083255 to TD

## Additional information

### Funding

| Funder | Grant reference number | Author |
| --- | --- | --- |
| National Institutes of Health | R01 AI083255 | Terje Dokland |

The funders had no role in study design, data collection and interpretation, or the decision to submit the work for publication.

### Author contributions

Altaira D Dearborn, Formal analysis, Validation, Investigation, Writing—review and editing; Erin A Wall, Formal analysis, Investigation, Methodology, Writing—review and editing; James L Kizziah, Data curation, Validation, Investigation, Visualization, Methodology, Writing—review and editing; Laura Klenow, Resources, Investigation, Methodology; Laura K Parker, Resources, Validation, Investigation, Methodology; Keith A Manning, Validation, Investigation, Methodology, Writing—review and editing; Michael S Spilman, John M Spear, Resources, Software, Methodology; Gail E Christie, Conceptualization, Supervision, Validation, Investigation, Methodology, Project administration, Writing—review and editing; Terje Dokland, Conceptualization, Supervision, Funding acquisition, Investigation, Visualization, Methodology, Writing—original draft, Project administration, Writing—review and editing

### Author ORCIDs

Terje Dokland http://orcid.org/0000-0001-5655-4123

### Decision letter and Author response

Decision letter https://doi.org/10.7554/eLife.30822.027
Author response https://doi.org/10.7554/eLife.30822.028

## Additional files

### Supplementary files

• Supplementary file 1. List of primers used to construct the plasmids in this study. Restriction sites are underlined.
DOI: https://doi.org/10.7554/eLife.30822.018

• Transparent reporting form
DOI: https://doi.org/10.7554/eLife.30822.019

### Major datasets

The following datasets were generated:

| Author(s) | Year | Dataset title | Dataset URL | Database, license, and accessibility information |
|---|---|---|---|---|
| Kizziah JL, Dearborn AD, Dokland T | 2017 | Capsid protein and C-terminal part of CpmB protein in the Staphylococcus aureus pathogenicity island 1 80alpha-derived procapsid | https://www.ebi.ac.uk/pdbe/entry/emdb/EMD-7035 | Publicly available at the EMData Bank (accession no. EMD-7035) |
| Kizziah JL, Dearborn AD, Dokland T | 2017 | Capsid protein and C-terminal part of scaffolding protein in the Staphylococcus aureus phage 80alpha procapsid | https://www.ebi.ac.uk/pdbe/entry/emdb/EMD-7030 | Publicly available at the EMData Bank (accession no. EMD-7030) |
| Kizziah JL, Dearborn AD, Dokland T | 2017 | Capsid protein and C-terminal part of scaffolding protein in the Staphylococcus aureus phage 80alpha procapsid | https://www.rcsb.org/pdb/explore/explore.do?structureId=6B0X | Publicly available at the RCSB Protein Data Bank (accession no. 6B0X) |
| Kizziah JL, Dearborn AD, Dokland T | 2017 | Capsid protein and C-terminal part of CpmB protein in the Staphylococcus aureus pathogenicity island 1 80alpha-derived procapsid | https://www.rcsb.org/pdb/explore/explore.do?structureId=6B23 | Publicly available at the RCSB Protein Data Bank (accession no. 6B23) |

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
