## [Decision Letter]

Thank you for submitting your article "Competing scaffolding proteins determine capsid size during mobilization of *Staphylococcus aureus* pathogenicity islands" for consideration by *eLife*. Your article has been favorably evaluated by Gisela Storz (Senior Editor) and three reviewers, one of whom is a member of our Board of Reviewing Editors. The reviewers have opted to remain anonymous.

The reviewers have discussed the reviews with one another and the Reviewing Editor has drafted this decision to help you prepare a revised submission.

Summary:

The article by Dearborn et al. titled "Competing scaffolding proteins determine capsid size during mobilization of Staphylococcus aureus pathogenicity islands" supplies high-resolution (i.e. near atomic) cryo-EM data of T=7 and T=4 icosahedral structures and provides mechanistic insight into the role of phage 80α scaffolding protein and the SaPI protein CpmB in regulating size determination during capsid assembly. Understanding how capsids are assembled and the factors that regulate these processes has been a long-standing debate in the field. Particularly mysterious has been the fundamental question of how scaffolding proteins direct assembly of capsids into different sizes and few structures are available showing direct scaffolding protein interactions with capsids. This manuscript is nicely written and provides insightful conclusions about how scaffolding proteins (or scaffolding protein mimics like CpmB) function to control T number. Both the cryo-EM data and the mutagenesis approaches are elegant. Strikingly, the local interactions between CpmB and SP with the capsid are nearly identical, but it appears that long-range interactions are responsible for constraining capsid shell diameters.

Essential revisions:

1) The work presented covers significant ground and is generally well supported and suitable for *eLife*. However, the basis for "atomic modeling" is the cryoEM density, and here the data presented include a handful of capsid images by cryoEM, surface views of the capsids, and one short segment of density demonstrating sidechain visualization (Figure 2). The latter is troubling as of the 9 residues indicated, only 4 show convincing density in which to position the sidechains while the other 5 are outside the density. For such a key aspect of the paper on which much else depends, the phrasing "many side chains were clearly resolved (Figure 2)" is not apparently accurate. The 5 density-free sidechains are identified, but no other reference is made to this panel and the purpose of their labeling is unclear. Perhaps the point has been missed, or perhaps the authors can show a more convincing region of density where the quality of sidechain visualization is better revealed, either way the basis for the atomic models presented (Figure 4, Figure 6) needs to be more strongly supported. The density quality may also be demonstrated by sections, perhaps in a supplemental figure.

2) No mention is made of structures being deposited in the EM Databank (cryoEM maps) or the PDB (models).

---

## [Author Response]

Essential revisions:1) The work presented covers significant ground and is generally well supported and suitable for eLife. However, the basis for "atomic modeling" is the cryoEM density, and here the data presented include a handful of capsid images by cryoEM, surface views of the capsids, and one short segment of density demonstrating sidechain visualization (Figure 2). The latter is troubling as of the 9 residues indicated, only 4 show convincing density in which to position the sidechains while the other 5 are outside the density. For such a key aspect of the paper on which much else depends, the phrasing "many side chains were clearly resolved (Figure 2)" is not apparently accurate. The 5 density-free sidechains are identified, but no other reference is made to this panel and the purpose of their labeling is unclear. Perhaps the point has been missed, or perhaps the authors can show a more convincing region of density where the quality of sidechain visualization is better revealed, either way the basis for the atomic models presented (Figure 4, Figure 6) needs to be more strongly supported. The density quality may also be demonstrated by sections, perhaps in a supplemental figure.

We have included a more extensive figure (Figure 3—figure supplement 1) showing the density on which the modeling was based, including one whole subunit of each structure (80a and SaPI1) and details of the density that we believe more accurately reflect the quality of the data. Obviously, at 3.8Å resolution, all side chains are not clearly resolved, and as others have reported, (negatively) charged side chains are especially prone to having poor density. Consequently, we do not expect all sidechains to be accurately modeled; however, the main chain should be correctly fitted throughout the structure, and the FSC curve between the model and the map shows that the model is accurate to the resolution of the map.

We have modified the text in the results, subsection “Structures of 80α and SaPI1 procapsids”, first paragraph, to more accurately describe the quality of the map that the model was based on.

2) No mention is made of structures being deposited in the EM Databank (cryoEM maps) or the PDB (models).

Submission to EMDB and PDB has now been completed, with accession numbers EMDB-7030, PDB ID 6B0X for 80a and EMDB-7035, PDB ID 6B23 for SaPI1. A statement to this effect has been added in the Materials and methods section, at the end of the subsection “Three-dimensional reconstruction and model building”.